

**A simplified system to quantify storage of carbon dioxide, water vapor and heat within a**
**maize canopy**
Taqi Raza[1*], Neal S Eash[1], Joel Nathaniel Oetting[1], Bruce B Hicks[2]
[1]Department of Biosystems Engineering and Soil Science, The University of Tennessee, Knoxville
USA
[2]MetCorps, Norris, USA
*Corresponding
Taqi Raza
taqiraza85@gmail.com, traza@vols.utk.edu
Department of Biosystems Engineering and Soil Science, The University of Tennessee, Knoxville
USA

27

28

29





**Highlights**

1. A new multiport profile system simplifies examining the $CO_2$ and $H_2O$ storage terms.

2. Neglecting canopy storage terms leads to inaccuracies in crop energy balance closure.

3. Energy closure is improved if canopy storage is considered.

4. The multiport system is necessary for eddy covariance-derived fluxes over maize canopy.

**Abstract**

The significance of canopy storage of $CO_2$, latent energy and sensible heat within agricultural crops has not been fully examined. Reported canopy storage terms are consistently smaller than found for a forest ecosystem, such that they are often neglected. A novel multiport profile system has been developed to examine these storage terms. The system sequentially samples air from four heights to a single non-dispersive Infrared Gas Analyzer (IRGA). Following extensive laboratory testing, the system was used to measure $CO_2$ and $H_2O$ within an eastern Tennessee maize canopy in 2023. The storage of latent and sensible heat was large enough to merit incorporation with conventional field measurements. The new system will enable profile measurements of $CO_2$ sufficient to quantify canopy storage terms as are needed in agricultural field campaigns.

**Keywords:** Multi-port system, vertical canopy profile, storage terms, energy balance closure, maize, carbon sequestration

## 1  Introduction

In the last few decades, significant work has been conducted to improve understanding of gaseous exchanges between soils, plants, and the atmosphere. These improvements have been rapidly incorporated into land-surface models and numerical-based weather prediction as well as assessment of atmospheric fluxes of carbon dioxide (Lamas Galdo et al., 2021), water vapor (Wang et al., 2023), and heat over vegetated landscapes (e.g., Tilden and Steven, 2004). Eddy covariance (EC) is a widely accepted method to measure the fluxes of $CO_2$, $H_2O$, and heat in the ecosystem (Nicolini et al., 2018). Routine EC measurements are now made at more than 650 locations, distributed globally (Fluxnet; Baldocchi, 2003).

At most forest experimental sites, the measured energy budget is not always close to the balance calculated by the conventional method expressed by Eq. (1) (Wilson et al., 2002). An important



factor emerging from forest ecosystem studies is that storage terms contribute substantially to
the energy closure of forests and to the quantification of evapotranspiration (McCaughy and
Saxton, 1988). In most forest studies, storage terms are ignored in consideration of the energy
balance equation:
$$R_n - G = H + LE \qquad (1)$$
Here, $R_n$ is net radiation, $G$ is soil heat flux, $H$ is sensible heat flux and $LE$ is latent heat flux.
Storage measurement is challenging due to temporal changes in $CO_2$, $H_2O$, and heat (Yang
et al., 1999). Globally, only a few sites (less than 30 %) apply a profile measurement system to
calculate the temporal variations and storage terms (Papale, 2006). Many studies have reported
that energy balance closure is an unsolved problem for a variety of vegetation types: the sum of
sensible and latent heat flux is found to be 10-30% lower than the available energy (Wilson et
al.,2002; Twine et al.,2000; Leuning et al. 2012; Russell et al. 2015; Liu et al. 2017; Raza et al.,
2023a). There are several possible reasons for energy closure errors resulting from EC
experimentation, such as neglecting the canopy and soil storage terms, loss of low- or high-
frequency flux components, and the use of inappropriate averaging times, etc. (Massman, 2000;
Meyers and Hollinger, 2004). Measurement procedures to test energy balance closure vary by
researchers and there is no standardized way to address the issues that arise.
In the case of agricultural cropping systems, storage terms are considered small and are
often ignored (Raza et al., 2024; Nicolini et al., 2018). Studies on the assessment of storage terms
within agricultural ecosystems are few, but the matter is well documented by researchers in the
case of forest ecosystems studies (Mayocchi and Bristow, 1995; Wilson et al., 2002).
Storage terms quantification is challenging because of its requirement for measurements
both within and above the canopy (Yang et al., 1999). Finnigan (2006) reported that the storage
term is underestimated when the average sampling time is large. Neglecting canopy storage
terms in studies of Net Ecosystem Exchange (NEE) can also cause substantial errors (Raza et al.,
2023b). To understand the role of the storage terms in energy balance closure and NEE, new
measurement and analysis approaches are required (Irmak et al, 2014).
In the most recent series of field experiments conducted by the present research team,
as now reported, the emphasis has also been on fluxes of carbon dioxide ($CO_2$). The field site is



large enough to warrant the use of EC measurement systems without fears of fetch and/or
footprint limitations (q.v., Foken et al., 2017). Measurement procedures followed the
recommendations of FLUXNET (Wilson et al., 2002) and ICOS (Montagnani et al., 2018). A key aspect
of the research program was the requirement for $CO_2$ concentration measurements at several
heights within the plant canopy, to permit examination of (1) flux interactions with pooled $CO_2$
at night; (2) the $CO_2$ storage term derived from EC observations; and (3) sub-canopy mixing.
The purpose of the present paper is to describe a measurement system specifically
designed to provide observations for assessing the quantities contributing to the diurnal heat
cycle, including the various storage terms and net ecosystem exchange. The instrumentation now
described will be used to extend the analysis into the inter-canopy airspace, using eddy
covariance observations as a basis for assessing storage terms. The protocols recommended by
the ICOS community (e.g., Montagnani et al., 2018) have been used as guidance.
**2  Methodology/Configuration**
**2.1 Apparatus design, operation, and measurement**
Field experiments have repeatedly shown that the need for an uninterrupted series of
observations of difficult–to–measure variables impose technical requirements that are often
difficult to satisfy. In practice, the more complicated a measurement system, the more likely the
desired continuous records will be interrupted by instrumentation malfunctions or by a variety
of other unanticipated issues. Such interruptions have been a challenge in the many studies of
maize crops conducted by the University of Tennessee (O'Dell et al, 2014; Hicks et al., 2020) in
their series of field experiments These field experiments (in Lesotho and Zimbabwe as well as in
Tennessee and Ohio in the USA) have demonstrated the need for a reliable yet technically simple
measurement system to measure profiles of the quantities of interest, within and above a
growing crop. To satisfy the basic requirements for time continuity and reliability of the data
record, a multi-port sampling system has been developed. The intent is to facilitate the routine
acquisition of temperature, humidity, and carbon dioxide data within and above a maize canopy.
Analysis of the recorded observations requires attention to gradients of the variables measured
and well as to the variables themselves. To minimize consequences of individual sensor offsets
when gradients are computed, the new system is designed to use a single detection system (an





IRGA —LI-COR-850, $CO_2$/$H_2O$ gas analyzer). In the application considered here, the system was
used to measure four heights, two within a maize canopy and two above.
Figure 1 presents a schematic description of the apparatus. The system is designed to maintain
continuous airflow through all intake tubes, to cycle through all heights of measurement every
minute and to minimize the switching time between samplings. The system consists of two small
pumps [Model TD-3LSA, Brailsford & CO., INC. Antrium. NH, USA], one pump (purge pump) draws
in the sampling air to maintain constant flow to minimize hygroscopic interactions along the tube
wall while the other pump pushes the drawn air to the IRGA. The sampling pump is mounted
close to the IRGA so that air smoothly enters the IRGA at ambient pressure. The airflow rate
through the sampling tubes is regulated by a flow meter [LZQ-7 flowmeter, 101.3 KPa, Hilitland]
at 700 ml/min; the flow rate through the IRGA is maintained 1000 ml/min. The switching between
heights is controlled by four three-way solenoid valves [231Y-6, Ronkonkoma, NY, USA]. The body
material of the solenoid is brass, and the internal component material is stainless steel as is
required when water vapor is present.
Each sampling tube is 10.5 m long to ensure each sampling height has the same transit time. The
purge pump manifold and all sampling tubes are of the same kind of urethane [BEV-A-LINE,
Polyethylene material, Cole Parmer]. Before passing through the analyzer, the air is passed
through a 1-µm pore filter [LI-6262, LI-COR NE, USA] to avoid the drifting of the analyzer and
pumps during the time of measurement due to the accumulation of debris, dirt, particulates, etc.,
that can cause blockage in the analyzer optical cells. The air outlet of the purge pump and IRGA
are open directly to the atmosphere.










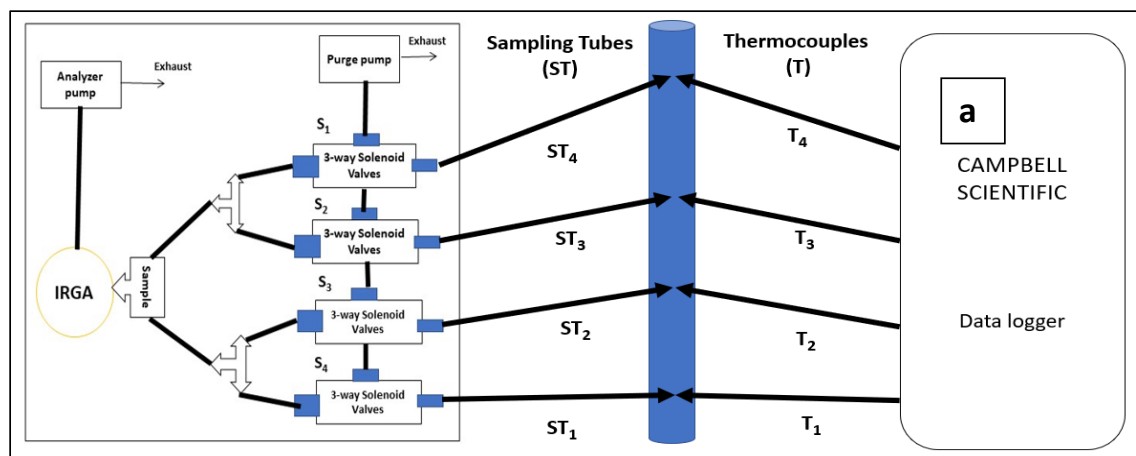

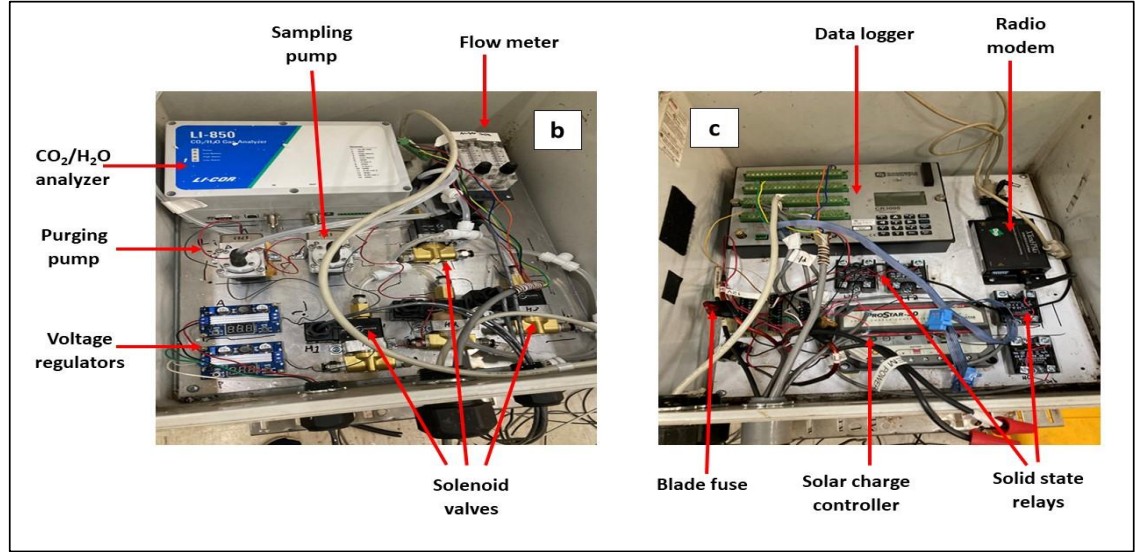

**Figure 1abc** (a) Schematic diagram of the manifold for profile sampling of $CO_2$, water vapor ($H_2O$), (b) analyzer, pump, and manifold system, (c) data logger for data collection

**2.2 Sampling time**
To determine the sampling time of the multiport system for accurate measurement of $CO_2$ and
$H_2O$, the system was first evaluated in the laboratory. The apparatus was first flushed with
nitrogen ($N_2$) gas to create a zero-carbon dioxide (0 ppm) environment. Subsequently, a known



concentration of $CO_2$ (430 ppm) at ambient pressure was fed through the intake tubes and
system outputs were measured, with results as shown in Table 1 and Fig. 2. This process allowed
determination of the minimum amount of time reach a stable measurement reading.
To derive a continuous record of concentrations at each of the heights of interest (in the present
experiment, four of them) switching between heights was set at every 7.5 seconds allowing each
of the heights to be sampled twice a minute. Figure 2 shows the delays associated with the
switching; these are confirmed by consideration of the known travel length and flow rate in the
tubes. The delay (3.2 seconds) in reading by the IRGA was due to the presence of residual air in
the previous sampling tube and other components of the apparatus, including the solenoid and
manifold (refer to Fig. 1). This delay indicates how much time the system takes to purge the
shared air in the manifold system. The sampling pump has a flow rate of 1 L/min, optimized to
maximize cycling time and minimize any water vapor surface interaction in the urethane tubes.
3.2 seconds were ignored, and the remaining 4.3 seconds were recorded by the datalogger.
The laboratory tests showed that as the IRGA received known [$CO_2$], it took approximately
1.8 seconds to achieve a steady output. During the laboratory evaluation period, the recorded
error was less than 0.5% in [$CO_2$] between sampling heights as shown in Table 1. An accuracy
error of less than 1% is well within the acceptable range for the IRGA now used, according to the
specifications provided by the manufacturer. Montagnani et al. (2012) found 11% error for a set
of measurements when estimating $CO_2$ storage flux using the ICOS method.

**Table 1** IRGA output for multiport air sampling system for $CO_2$ conc. (430 ppm) fed through the
sampling manifold at ambient pressure.

| Sampling tube | Mean CO₂ concentration (ppm) | SD | No. of samples | Error % |
|---|---|---|---|---|
| Intake 1 | 430 | 0.474 | 23 | 0.00 |
| Intake 2 | 431 | 0.196 | 15 | 0.20 |
| Intake 3 | 431 | 0.167 | 16 | 0.20 |
| Intake 4 | 432 | 0.119 | 13 | 0.46 |





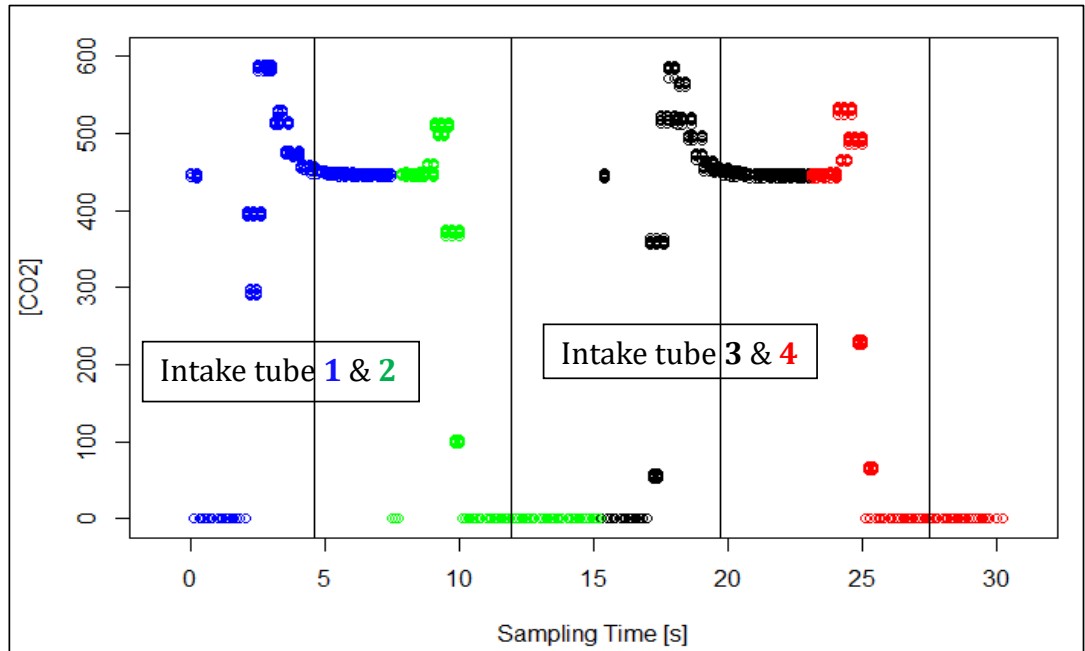



**Figure 2** The time-dependent relationship between the infrared gas analyzer (IRGA) in the multiport air sampling apparatus for a gas concentration of 430 ppm $CO_2$ flowing at <1 L/min. The switching between the intakes occurs every 7.5 seconds. Blue corresponds to sampling intake height 1, green to height 2, black to height 3, and red to height 4. The vertical lines demarcate the stable, equilibrium regions where the measurements were suitable to be recorded. Here the lines were at 4.9 seconds but were further improved to 3.2 seconds.

### 3 Field measurement setup

Following laboratory testing, the system was deployed in a field study conducted at Loudon, Tennessee, in 2023. In this study, four intake sampling tubes were positioned at heights (m) of 0.11, 0.5h, 1+h, and 2+h, where h is maize canopy height above the soil surface. Note that one height was permanently set at 0.11 m and three of these heights were adjusted as the crop grew. Sampling intakes were positioned on a 10 m steel mast at the respective positions. Thermocouples at the same height were used to measure temperature within and above the



canopy; these thermocouples were aspirated within a white PVC pipe of 1.9 cm diameter (Figure
3) that also served as a radiation shield.

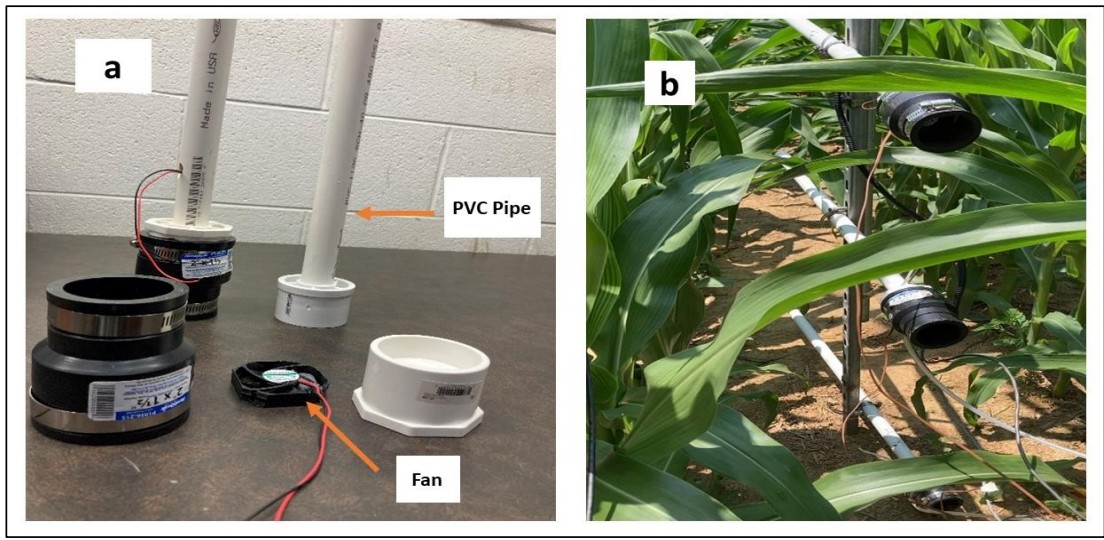


**Figure 3ab** Aspiration of $CO_2$ intake tubes and thermocouples and their application in the maize
canopy.
To provide the measurements necessary to interpret the gradient observations, a tripod tower
was used to support an eddy covariance system and supporting micrometeorological variables
— an IRGASON [$CO_2$/$H_2O$] open path gas analyzer system, [Campbell Scientific, Logan, Utah], a
net radiometer [Kipp & Zonen SR# 103660, OTT HydroMet B.V. Delft, Netherland], infrared
radiometers [IRs-S1-111-SS, Apogee Instruments Inc, USA], and type T thermocouples [Omega,
USA]. A schematic diagram of the system is shown in Fig. 1. The system was visually inspected
every week for any leakage, condensation, and contamination.
**3.1 Experimental site**
The field study was conducted near Philadelphia, in Loudon County Tennessee (35.6729° N,
84.4651° W). The study area is twenty-three hectares of agricultural farmland cultivated with a
maize cropping system. The red point on the map represents the location of the site where the
system was installed. The mean annual temperature and precipitation of the site are 12-15 ºC
and 132-142 cm respectively. The elevation and slope of the site are 280 m (Figure 4A) and 2-5%



(Figure 4B) respectively. The soil was classified as Alcoa Loam (Rhodic Paleudult) according to the
USDA classification scheme (Soil Taxonomy, 1976).

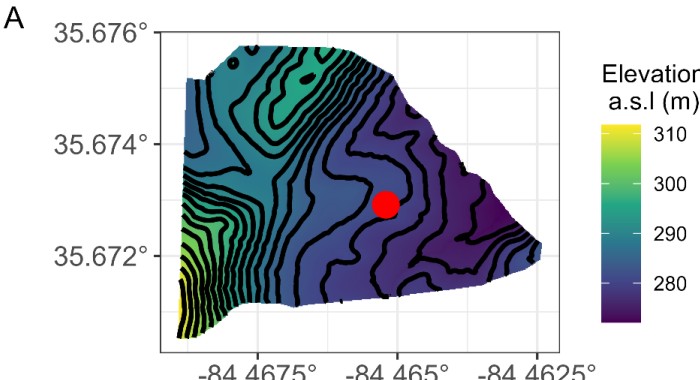

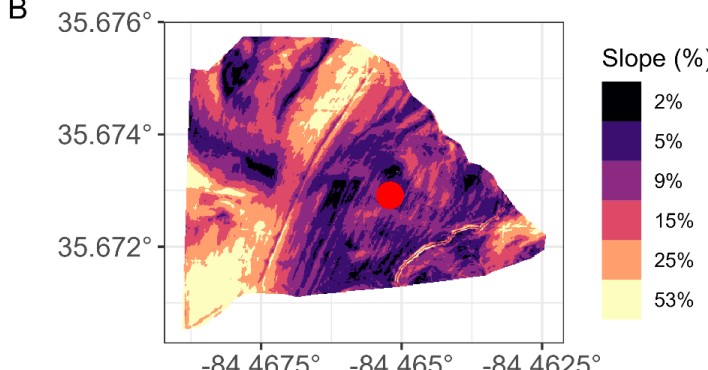


**Figure 4** Soil elevation and slop overview of the test site in rural Philadelphia Tennessee, USA.
A) The red point on the map shows the instrumentation location.
**3.2 Calculation of storage terms**
In accordance with other studies of the surface energy budget using EC systems, storage terms refer to
depletion or accumulation of scalar quantities ($CO_2$, $H_2O$, etc.) in a hypothetical control volume beneath
the height of EC flux measurement. A storage flux is defined as the rate of change of dry molar
concentrations of the same variables within the same control volume. Both concepts relate most directly
to the conditions of "perfect" micrometeorology. In practice, natural complexities of surroundings and



exposures interfere to the extent that will necessarily be site-specific. Moreover, the covariances that
are central in related deliberations are statistical quantities, with well-recognized error margins
associated with every quantification of them. During this study, the storage fluxes of scalar quantities
($CO_2$, water vapor, etc.) were calculated using the ICOS methodology (Montagnani et al., 2018). For the
case of CO2,

$$J_c = \overline{\rho_d} \sum_{i=1}^{N} \left( \frac{\Delta c}{\Delta t} \right)_i \Delta z_i.$$    (2)


Here, $J_c$ is the storage term of $CO_2$ within the $i_{th}$ layer over which $\Delta c$ is measured, $\Delta z_i$ is the
thickness of this layer and $\Delta t$ is the measurement time step; $\overline{\rho}_d$ is dry air density, and N is the
number of layers.
**4   Results and discussion**
**4.1 Vertical profile of $CO_2$ within a maize canopy**
Previous experiments revealed the ubiquity of nocturnal pooling of $CO_2$ because the presence of
the maize canopy and the development of a strongly stable atmospheric surface layer that
permits $CO_2$ emitted by the soil biota to accumulate overnight. Fig. 5 presents average diurnal
cycles of $CO_2$ concentrations measured at four heights, two within the canopy and two above.
The concentrations of $CO_2$ observed low in the canopy exceed those elsewhere. Moreover, note
that the increasing concentrations within the pool closely parallel each other, providing support
for the assumptions made elsewhere about $CO_2$ profile linearity within the pools. The fact that
concentrations drop below ambient (~425 ppm) suggests that photosynthesis is ongoing which
rapidly reduces the canopy concentrations.
Following dawn (or sunrise as indicated in Fig. 5) the accumulated concentrations of $CO_2$ drop
rapidly as convection starts to mix surface air with that aloft and as photosynthesis commences.
Concentrations decrease to about 350 ppm, due to the maize photosynthetic requirement for
$CO_2$ from the air. After sunset, concentrations of $CO_2$ rise uniformly with little evidence of a
separation between sub-canopy and upper-canopy concentration regimes. Note, however, that
there is evidence of early effects of soil emissions, such that the 11–cm trend departs from the
others soon after solar noon.





Furthermore, at all four heights, $CO_2$ concentrations were greatest during late night and early
morning until 0600 local time (LT), after which concentrations declined rapidly and reached a
relatively constant level of approximately 350 ppm in the afternoon (1200 to 1800 LT). The 350
ppm observation is 70 ppm less than current ambient $CO_2$ due to photosynthetic demand.
Subsequently, $CO_2$ concentrations increased again, with a more pronounced increase during late
night. Soil respiration, photosynthesis, and temperature contribute to this trend. During the late
night, the surface atmosphere stabilizes and wind speeds decrease allowing emitted $CO_2$ to
accumulate. Moreover, in many climatic regions like our experimental site, nighttime soil
temperatures remain high enough to sustain microbial and soil respiration activities, resulting in
$CO_2$ accumulation in the stratified air above the ground. As the sun rises, increased light
availability increases stomatal activities, leading to higher photosynthesis rates.

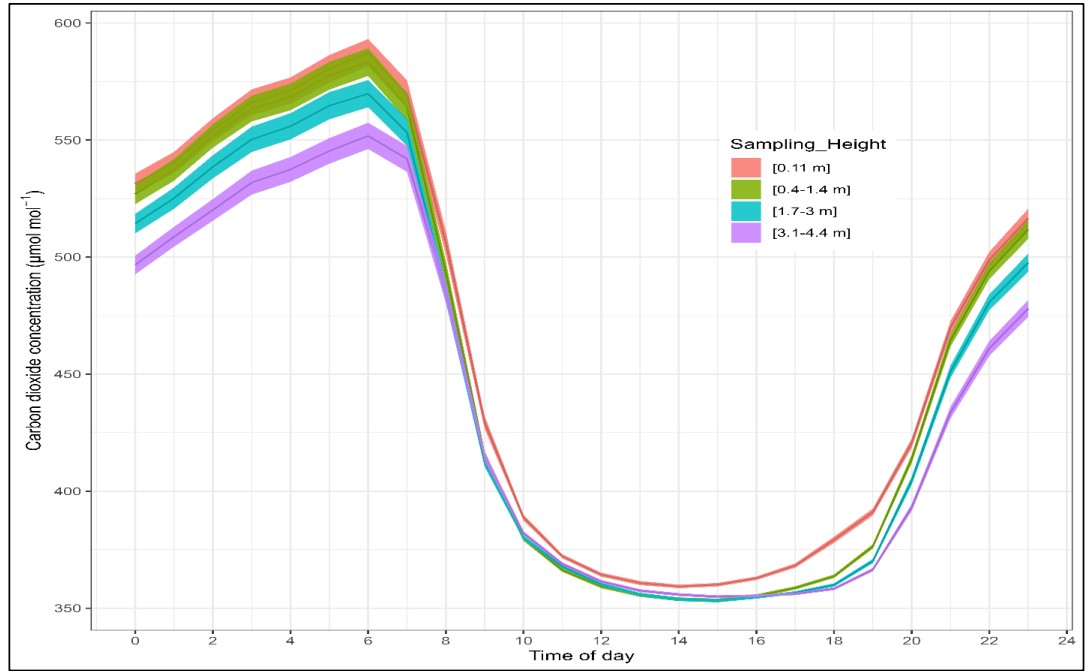

**Figure 5** Average diurnal vertical cycle of $CO_2$ for over two months of maize canopy
growth/monitoring. Times of sunrise and sunset are shown.
**4.2 Vertical profile of $H_2O$ in a maize canopy**
As in Fig. 5, Fig. 6 shows 15-minute $H_2O$ concentration observations have been used to construct
an average diurnal cycle for the two-month period exemplified shown here. During daytime, $H_2O$



concentrations were significantly higher when compared to nighttime, peaking between 1200 LT and 1400 LT and then gradually decreasing. Notably, after 2000 LT, we recorded a rapid decline in $H_2O$ concentration. At 0600 LT, the $H_2O$ reached its minimum concentration throughout the cycle, followed by a sharp increase in the first hour. The $H_2O$ concentration decreased as the height increased for both day and nighttime because at both times a source of water vapor is the soil surface, with crop evapotranspiration adding in the daytime.

A comparison of the diurnal cycles shown in Figs. 5 and 6 indicates considerably different cycles of $CO_2$ and $H_2O$ cases. At night, Fig. 5 shows that the $CO_2$ profile appears to be stronger than in the daytime. The opposite is seen, for $H_2O$, in Fig. 6. The reason is presumed to be that $CO_2$ continues to be emitted from the soil at night, whereas there is no parallel process influencing the $H_2O$ concentrations. This is a feature made apparent by the profile sampling system.

The processes of evaporation from the soil surface and evapotranspiration from leaves are linked with solar radiation. Overall, the study highlights the vertical distribution of water vapor concentration and its temporal variability, indicating that factors such as height and diurnal variations significantly influence the profile/gradient and temporal patterns of water vapor in the canopy profile.

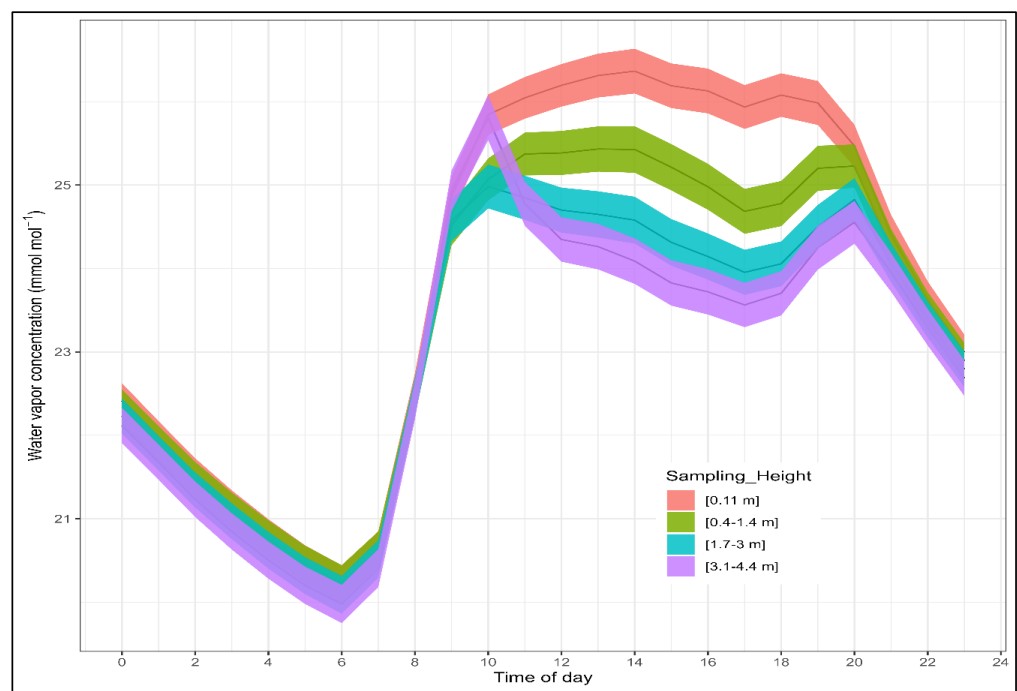

281

282

**Figure 6** Diurnal vertical profile pattern of water vapor averaged over two months as in Fig. 5.

**4.3 Latent heat, sensible heat and $CO_2$ storage fluxes of maize profile**

The vertical profile data were also used to investigate how various storage fluxes influence the energy balance closure of the maize crop. The diurnal average behavior of these storage fluxes is shown in Fig. 7. $CO_2$ storage (Fig. 7a) exhibited higher values at nighttime as compared to daytime, due to the $CO_2$ pooling effect. During both early morning and late night, $CO_2$ storage increased at a rate of approximately 1 $\mu$mol m$^{-2}$ s$^{-1}$, after which it gradually decreased until 0730-0800 LT when it became negative. $CO_2$ storage started to increase again, remaining close to zero until the following day. This trend indicates that $CO_2$ storage was significantly higher during early morning and late night as compared to daytime when photosynthesis processes actively utilized $CO_2$ within the canopy. During nighttime and morning, these processes were reversed, leading to $CO_2$ storage.

Sensible heat energy storage (Fig. 7b) was found lower than latent energy storage (Fig. 7c).





The diurnal patterns of sensible (Fig. 7b) and latent energy storage (Fig. 7c) show similar
characteristics. Sensible heat storage remained zero until after sunrise, eventually rising to a
maximum value (around 2.5 W m$^{-2}$) recorded between 1200 LT and 1230 LT. After that, this
sensible energy storage rate declined, reaching negative values until 2400 LT and returning to
zero until 0700 LT the following day.
Latent energy storage (Fig. 7c) exhibited a pattern like sensible energy storage but with
comparatively higher values. The maximum latent energy storage (> 4 W m$^{-2}$) occurred between
0700 LT and 0730 LT, followed by a rapid decrease and negative storage until 2000 UTC. After a
brief increase (presently unexplained, about 4 W m$^{-2}$) for thirty minutes, rapid decline ensued,
leading to negligible values during the late nighttime until the next morning. The diurnal
variations in sensible and latent energy storage are influenced by several factors, including solar
radiation, temperature fluctuations, and plant physiological processes. The variation in canopy
density, structure, and microclimate within the canopy significantly changes the canopy storage
which directly influences the daily integrated fluxes. Forest studies reported that the exchange
of CO2 and H2O between plant and atmosphere is regulated by canopy density (e.g. LAI), surface
conductance, etc. and these canopy characteristics change seasonally which influences the
estimation of flux (Renchon et al., 2024; Chen et al., 2019). Therefore, consideration of canopy
storage into our analysis can improve the accurate estimation of flux.
The results provide valuable insights into the energy dynamics within the maize canopy and
contribute to a deeper understanding of its environmental response to plants during different
periods of the day. Latent and sensible heat storage are the two most important components of
the energy balance of maize crops and are primarily influenced by important environmental
conditions such as temperature, humidity, solar radiation, and wind.




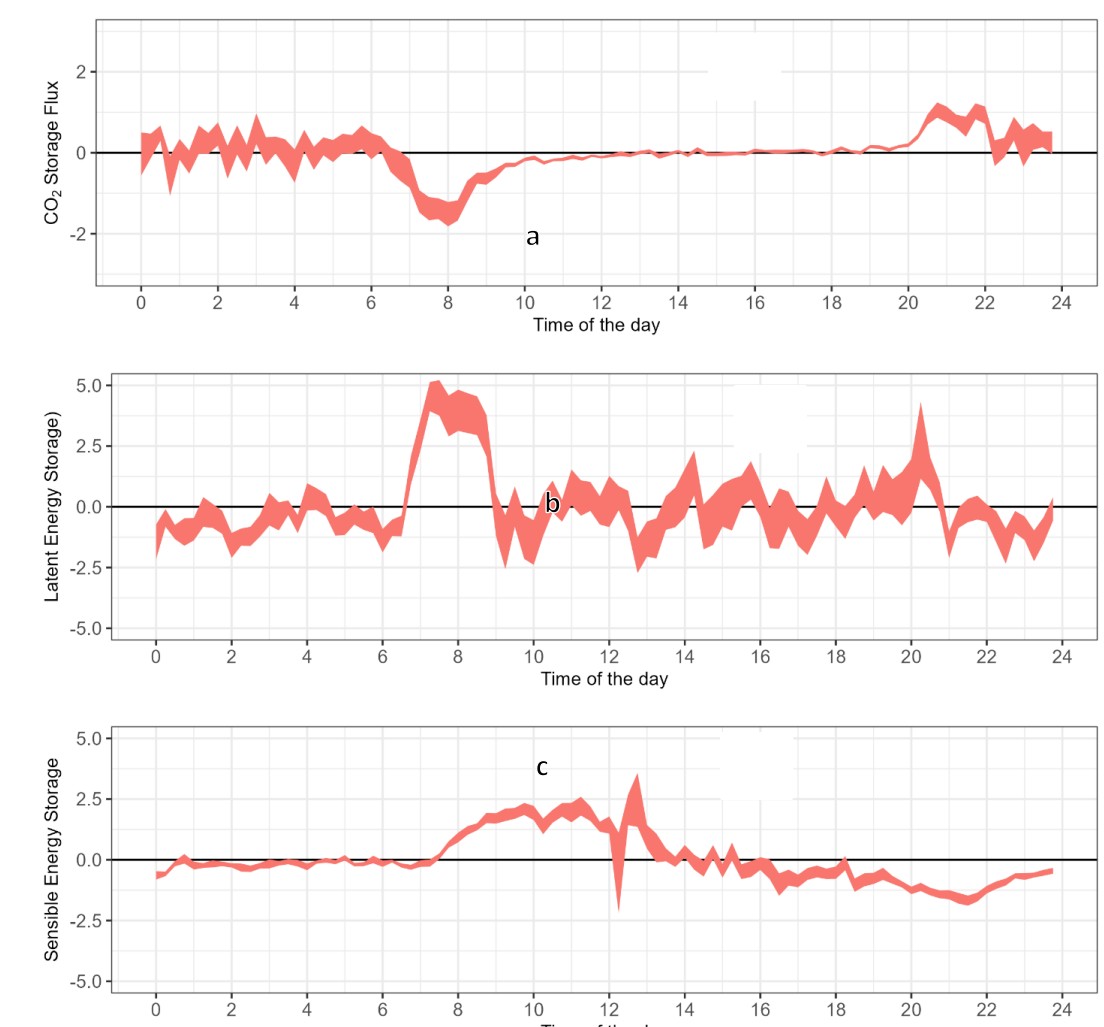



**Figure 7abc** Diurnal pattern of $CO_2$ storge (µmol m$^{-2}$ s$^{-1}$), sensible energy storage (J m$^{-2}$ s$^{-1}$) and latent energy storage (J m$^{-2}$ s$^{-1}$) of a mature maize crop, averaged over a two month period.

Note that the definition of the heat storage used here (as in Eq. (2)) omits warming of the biomass. This omission accounts for the differences between the storage terms now computed and those published previously (e.g. Hicks et al., 2020), based on infrared measurements of



biomass temperature in addition to changes in air temperature and humidity below the level of atmospheric measurement.

**5   Conclusions and summary**

The new multi-port profile system demonstrated its effectiveness in the measurement of $CO_2$ and $H_2O$ concentrations at different heights. Its development significantly aids in understanding $CO_2$ and $H_2O$ concentration variations in the vertical profile of a rapidly growing maize crop, thereby facilitating precise assessments of their exchanges, storage, and overall balance within agricultural ecosystems. The new system is designed to provide the capability to change measurement heights simply, as crops grow in height, while relying on a single measurement device and thereby minimizing level-to-level biases.

The 2023 field experience with the new system indicates that canopy data obtained from the vertical profile observations  hold potential for many  applications in future studies such as evaluation of soil-plant-atmospheric models that rely on the precise estimation of $CO_2$, heat, and $H_2O$ concentrations as well as future assessment of canopy nitrous oxide concentration.

**Author contribution statement**

**TR:** Data curation, Formal analysis, Methodology, Visualization, Writing – original draft. **NSE:** Supervising, Funding acquisition, Project administration, Writing – review & editing. **JNO:** Formal analysis, writing and reviewing. **BBH:** Supervision, Writing – review & editing.

**Funding**

This work was supported by DuPont Tate & Lyle Bio Products Company.

**Declaration of computation intertest**

Authors declare no competing interest associated with this submission.

**Acknowledgment**

This work was supported by the University of Tennessee, Knoxville. The authors are very thankful to David R Smith (Senior Technical Specialist, BESS, UTK), Wesley C Wright (Senior Research Associate, BESS, UTK), Scott Karas Trucker (Senior Technical Specialist, BESS, UTK) and Josh Watson (Farmer) for their continuous support during this work.

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
