# Peer review of "A simplified system to quantify storage of carbon dioxide, water vapor and heat within a maize canopy"

_EGUsphere, 2024_

## Referee Comment (RC1)

**Manuscript ID:** egusphere-2024-2041

**Verdict:** Mandatory Major Revision

**Article Summary and General Remarks:** The manuscript entitled "A simplified system to quantify storage of carbon dioxide, water vapor and heat within a maize canopy" highlights a novel set-up to quantify the surface energy energy budget, including storage fluxes, using a vertical array of inlets (2 within the canopy and above the canopy) that measures $H_2O$ and $CO_2$ sequentially in a 30 second cycle (or a minute—details about cycling are conflicted in the article (see below)). The aim is to illustrate the utility of the set-up and to highlight the importance of terms often ignored within the energy budget closure at the surface. The introduction lays out the previous work well leading up to the 2023 experiment in Tennessee. The Methodology/Configuration discusses the instrument set-up and measurements conducted reasonably well; however, there are some areas in section 2 that need explanation and the discussion of Figure 2 and how its presented, especially with respect to the vertical line references, is confusing. Section 3 discusses the experimental set-up. In this section there is text dedicated to the Experimental site and the calculation of storage fluxes. The calculation of storage fluxes should be moved to methodology since the authors are talking about a method they are using to calculate storage fluxes. There is no discussion in Section 3 about instances of precipitation, the diurnal wind structure, etc., which would be important for providing context about the weather variability during the two-month period. It highly recommended that the authors describe the weather conditions over this two-month period, and, if possible, subsample days without precipitation to understand impacts on the canopy gradient structure. Section 4 goes over the results and discussion of results. A subsection is dedicated to $CO_2$ storage, $H_2O$ storage, and the budget. While the first subsection is written reasonably well, the other two sections fall short of a rigorous analysis of the results with a flawed interpretation (see examples below). Furthermore, if the authors are interested in promoting the novel instrument design, then this paper ought to delve deeper into the measurement capability rather than provide just diurnal plots with limited interpretation of the results (see below). Finally, the conclusions in the final section does not summarize the key findings, shortcomings in the method, or a path forward to integrate the novel set-up as a mainstream measurement option. Below are the key highlights of the study laid out by the authors:

1. A new multiport profile system simplifies examining the $CO_2$ and $H_2O$ storage terms
2. Neglecting canopy storage terms leads to inaccuracies in crop energy balance closure
3. Energy closure is improved if canopy storage is considered
4. The multiport system is necessary for eddy covariance-derived fluxes over maize canopy

It is clear that the first two highlights are sufficiently covered, but the second two points are not adequately addressed in the present form of the manuscript. For instance, as a reviewer, I do not have an impression of an improvement. I see results highlighting the vertical structure of $CO_2$ and $CO_2$ storage, but not a clean budget calculation or comparison with and without $CO_2$ storage. Given the data available to the authors, I believe this should be possible. The last point is also a concern. EC is conducted with a fast response wind and temperature/tracer instrument on the order of 10Hz. The storage terms are not calculated as done with EC techniques and instead support the EC calculations of other terms that make up the surface budget.

In light of the comments above, I recommend a mandatory major revision before the manuscript is eligible for publication.

**Major Comments:**

[Figure]

Caption

Figure 2: In the caption, the authors note that the vertical lines represent stable periods where measurements are suitable for recording. The vertical lines and how they are placed in the figure does not seem to support the text in lines 148-151. The text suggests that feeding in a known amount of CO2 (430 ppm) and looking stability of these measurements define the periods of acceptable data. Based on Figure 1, intake tubes 1 and 2 (3 and 4) are coupled with a smooth transition from one intake tube to the other, while rapid increases and decreases occur as the measurement sequence switches from 2 to 3 and 4 to 1. If useful data is defined by obtaining the ambient CO2 for this test example, then the stability regions in the plot should be represented by time periods between the vertical solid and dashed lines in the annotated Figure 2 plot shared below. If this is not the case, then the authors need to improve the explanation of this plot, because it would seem undesirable to isolate and use the measurements when CO2 is 0 ppm as suggested in Figure 2 with the second and fourth vertical black lines. Lastly, why do measurement heights for tubes 2 and 4 record less data with respect to time compared to tubes 1 and 3 when measuring 430 ppm of CO2? I would think that achieving an approximately equal number of data points for each measurement height is preferred. Please address this.

Lines 179-181: If the authors were able to improve the set-up to measure at 3.2 seconds compared to 4.9 seconds, then the authors should show the plot with improved timing. The authors can show both figures with a panel dedicated to each that shows contrast. This is strongly encouraged.

Equation 2: The form of this equation uses two assumptions defined in Montagnani et. al. 2018: a homogeneous footprint and that the flux divergence can be ignored. While I have more confidence in the first assumption given that the survey region is a large maize field, I have concerns about neglecting the flux divergence term, especially during transitional periods (i.e., sunrise/sunset) where these calculations seem very important (e.g., Figures 5-7). If the goal is to minimize the error in the total surface budget, then how does neglecting the flux

divergence term contribute to the budget error? The authors should comment on this explicitly and note any caveats. It would also be a good idea to acknowledge this shortcoming and work towards evaluating these assumptions used when calculating storage terms.

Height of vertical measurements: The authors note that the height of measurements was adjusted with the growth cycle during the two-month period. The authors should present a table or a stacked bar graph showing height adjustments to measurements as a function of time so that readers know how often the heights were adjusted and the range of heights that were used in this analysis. Furthermore, it is recommended and strongly encouraged that the storage term calculations be conducted for each set of measurement heights to highlight how the storage terms evolved during the growth cycle. The results in their current form mask the growth cycle, which is no doubt an In important contribution to $CO_2$ and $H_2O$ storage, photosynthetic response and evapotranspiration. I suspect that changing the height of measurements does not change the result too appreciably given profile linearity. However, I wonder how the slope of profile linearity changed during the growth cycle. These kinds of analyses would strengthen the paper and support the approach and instrument set-up presented in the paper.

Shading in Figures 5-7: There is nothing in the text discussing the shading in these figures. I assume that the shading represents the variability within the two month period, but nothing is discussed about this shading. I do find it unlikely that there was almost no variability in $CO_2$ during the day over the two month period, and the that the results are almost flat (hovering around 350 ppm). The authors need to add a discussion explaining the shading, and it is strongly encouraged to go back into the data to examine variability at each half-hour averaging interval examined during the diurnal cycle.

Lines 288-290: The authors start out talking about morning and evening transitions, but only comment on the morning transition. This leads to a confusing interpretation if the reader views the statement made at the end of the sentence applicable to both morning and evening transitions when in fact it is not. Please comment on both and make the discussion on respective transitions clear.

Lines 295-297; 301-302: The authors claim that the sensible and latent fluxes exhibit similar characteristics. They do not.

Lines 302-303: The statement "…followed by a rapid decrease and negative storage until 2000 UTC" does not seem to support what is seen in Figure 7b. Sure, there was a sharp decline, but what followed was a fair amount of positive (accumulation) and negative (depletion) storage between this time period that does not lend support to this statement. The variability was large. Please revise messaging.

Lines 304: I believe the authors can avoid the statement "presently unexplained" by studying Equation (2) and and examining Figure 6. This is where the slopes in the profile collapse and an inflection is observed (i.e., concentrations begin to decrease). So the time rate of change of concentrations using the discretized form in Equation (2) should help you form a hypothesis, especially since this is during a period where the stable boundary layer forms. It should also be noted that while the peak in Figure 7b stands out during 2000 UTC, it is not considerably larger than some of the peaks and troughs between 900 UTC and 2000 UTC that reach magnitudes close to 2.5 W/m2.

Lines 303-305: Negligible? Visually I see statistically negative values during the night. The statistically negative latent storage (or depletion) follows a drop in water vapor concentration. The mechanisms for evapotranspiration are minimized and the uptake of moisture into the atmosphere near the surface is reversed as temperatures cool and condense at the surface.

The authors sort of allude to this in lines 272-277, but fail to link these results when discussing Figure 7.

Lines 301-313: Discussion in this paragraph is filled with erroneous statements and broad remarks. Dive deeper into the analysis of this figure and make the interpretation clearer and stronger.

Conclusions: The key findings are not summarized nor are caveats and shortcomings of the approach laid out. The conclusions would also benefit from statements related to future work.

**Minor Comments:**

Line 49: Insert "the" between "improve" and "understanding"

Line 56: can remove ", distributed"

Lines 72-73: Comment on "loss of low- or high-frequency flux components": What does this mean exactly? Was a filter applied? Resolution limitation? Was it the technique used? Please elaborate and add a reference if possible

Line 73: Remove "etc."

Lines 80-81: Perhaps instead say "Quantifying storage terms is challenging because measurements are required within and above the canopy". The current sentence is awkwardly phrased.

Lines 107-108: remove "in their series of field experiments"

Line 110: Remove comma between "interest" and "within". There are a lot of other examples of misplaced commas.

Line 115: replaced "and" with "as" before "well".

Lines 120-121 and Lines 154-155: In the first set of lines the authors remark on cycling through all heights every minute while the second set of lines indicates every 30 seconds. Which is it?

Line 195: change "variables" to "measurements"

Line 204: Add "in Figure 4" between "map" and "represents"

Lines 206-207: How did the authors account for the slope of terrain on sonic anemometer wind measurements since the plot indicates the circle is in 5-9% grade? Was the sonic anemometer orientated horizontally or was post processing done?

Line 207: How does the soil type factor into interpreting the results in this analysis? Soil type is important for moisture retention, capacity, and subsurface transport.

Line 218: move "be" between "necessarily" and "site-specific" before "necessarily"

Line 226:  Be clear about $\Delta z_i$.  Is it the separation distance between measurements?  If so, then the thickness for the bottom measurement would be between the surface and the measurement height (0.11 m).

Line 227:  I'm guessing that "N" equals 4, but this not stated explicitly.

Line 237:  "assumptions made elsewhere" should be backed up by citations.

Line 241:  "with that aloft"?  Or "with air aloft" or "with the overlying atmosphere"

Line 245:  can replace ", such that" with "as"

Paragraphs in lines 240-246 and 247-257 share a lot of redundant information

Lines 252:  Question on the sentence beginning with "Soil respiration".  Are you referring to the entire diurnal trend, day portion or night portion?  The sentence that preceded this one focused on the pattern at night, and as such, leads to confusion when interpreting the "Soil respiration" sentence.

Line 253:  It is remarked that wind speeds decreased at night.  Statistically, this may be true, but moderate to strong winds can develop at night which would effect the results and add to the variability observed and not discussed.  At a minimum the authors should note that they confirmed the diurnal wind structure.  They wouldn't need to show it, just make it clear that this was observed during the two-month survey period.

The authors never discuss the time range that this study was conducted.  Please indicate the months that this study took place before introducing Figures 5-7.  This should be discussed in Section 3.

Line 265:  Replace ", we recorded" with "is"

Line 268:  "height increased". The height of what increased?

Line 271:  The "profile appears to be *stronger* than" is not an appropriate description of the profile.   I think what the authors meant to say is that the profile is steeper or that there is a more pronounced vertical gradient.

Line 271:  Remove "cases" after H2O

Line 272:  Remove commas between "for H2O"

Figure 6 caption:  You can omit "pattern" after "profile"

Line 286:  Change "diurnal average" to "average diurnal".

Line 287:  Change "higher values" to "larger magnitudes and more variation".  Higher values could mean more positive.  As such, I would instead go with the suggested change.  Can also remove "as" between "nighttime" and "compared".

Line 295:  Sensible heat energy storage was not always lower than latent heat storage.  Also, references to "7b" and "7c" are incorrectly placed here and elsewhere in this section.

Lines 293-294:  Comment on "During nighttime and morning, these processes were revered, leading to CO2 storage".  During the morning transition the CO2 storage becomes negative (depletion) while during the night storage increases (accumulation).  Make this clear instead of just saying "leading to CO2 storage".

Missed opportunity connecting Figures 7a and 7b:  The authors should dig into the inflected results between CO2 storage (7a) and latent energy storage (7b) that occur between 0630 UTC and 0900 UTC during the hours following sunrise.  Note the importance of both CO2 and water to photosynthesis which activates at the time where the inflected behavior is observed.

Line 298:  change "After that, this" to "Afterwards the"

Lines 299-300:  "returning nearly to zero". Its not actually zero

Line 305:  Insert "afternoon into" between "the" and "late nighttime"
Figure 7:  There is not a comment about how the data is being processed to form the plots.  Is the data vertically integrated?  I'm guessing not given the units.  If so, what height are the authors choosing or is vertical averaging done?  Y-axis label in Figure 7b has a parenthesis that shouldn't be there.  Please place the a-c labeling in a corner and not over the plots.

Line 313:  Change from "the accurate estimation of flux" to "the accurate estimation of fluxes"

Figure 7a:  How do you explain the large variation between negative and positive CO2 at night and virtually no storage/no variability during the day?

Figure quality needs general improvement.  Larger fonts, particularly in Figures 5-7 is needed.

---

## Author Comment (AC1)

**Date:** Thursday, September 26, 2024

To,

Russel Dickerson
Handling Editor
**Atmospheric Measurement Techniques**

Sub.:- Submission revision RC1 of research paper to the atmospheric measurement techniques

Dear Sir,

Greetings of the day, I hope you are doing well! With reference to the above-cited subject, we are resubmitting herewith the revised version of research paper on **"A simplified system to quantify carbon dioxide, water vapor and heat within a maize canopy (egusphere-2024-2041)"** to your esteemed journal **"atmospheric measurement techniques".** Authors are grateful to the reviewer for a careful and helpful analysis of our manuscript. Undoubtedly, due to the Reviewer`s comments, the manuscript has been significantly improved. All the reviewer`s comments are reasonable, and we have corrected the MS in accordance with the comments and suggestions

Thank you for receiving our manuscript and considering it for further processing. Authors appreciate your valuable time and look forward to your response.

Sincerely yours,

Corresponding author

Taqi Raza

taqiraza85@gmail.com, traza@vols.utk.edu

Department of Biosystems Engineering & Soil Science

University of Tennessee, Knoxville USA

**Major Comments**

Comments: Figure 2: In the caption, the authors note that the vertical lines represent stable periods where measurements are suitable for recording. The vertical lines and how they are placed in the figure does not seem to support the text in lines 148-151. The text suggests that feeding in a known amount of CO2 (430 ppm) and looking stability of these measurements define the periods of acceptable data. Based on Figure 1, intake tubes 1 and 2 (3 and 4) are coupled with a smooth transition from one intake tube to the other, while rapid increases and decreases occur as the measurement sequence switches from 2 to 3 and 4 to 1. If useful data is defined by obtaining the ambient CO2 for this test example, then the stability regions in the plot should be represented by time periods between the vertical solid and dashed lines in the annotated Figure 2 plot shared below. If this is not the case, then the authors need to improve the explanation of this plot, because it would seem undesirable to isolate and use the measurements when CO2 is 0 ppm as suggested in Figure 2 with the second and fourth vertical black lines. Lastly, why do measurement heights for tubes 2 and 4 record less data with respect to time compared to tubes 1 and 3 when measuring 430 ppm of CO2? I would think that achieving an approximately equal number of data points for each measurement height is preferred. Please address this.

- After careful consideration, we decided to remove Table 1 (IRGA output for the multiport air sampling system, for CO2 conc. (430 ppm) fed through the sampling lines at ambient pressure) and Figure 2 (The time-dependent relationship between the infrared gas analyzer (IRGA) in the multiport air sampling apparatus for a gas concentration of 430 ppm $CO_2$ flowing at <1 L/min) from the manuscript, as it raised additional questions and complexities not central to the primary objectives of our study. The initial purpose of this figure was to demonstrate that the instrument was accurately measuring a known $CO_2$ concentration (430 ppm) using different intake tubes, which was a verification step supporting the field trial described in the text that follows. In the revised manuscript, we have clarified that both $N_2$ (0 ppm $CO_2$) and $CO_2$ (430 ppm) were used to test the stability of the system and confirm that the analyzer and switching mechanism were functioning correctly. We believe that the removal of this figure enhances the clarity and focus of the manuscript.

Comments: Lines 179-181: If the authors were able to improve the set-up to measure at 3.2 seconds compared to 4.9 seconds, then the authors should show the plot with improved timing. The authors can show both figures with a panel dedicated to each that shows contrast. This is strongly encouraged.

- Thank you for bringing this to our attention. We made a mistake. In fact, actual timing for measurement was 3.1 seconds and 4.4 seconds were ignored during the $CO_2$ reading by the analyzer. This has been briefly explained in the revised manuscript.

Equation 2: The form of this equation uses two assumptions defined in Montagnani et. al. 2018: a homogeneous footprint and that the flux divergence can be ignored. While I have more confidence in the first assumption given that the survey region is a large maize field, I have concerns about neglecting the flux divergence term, especially during transitional periods (i.e., sunrise/sunset) where these calculations seem very important (e.g., Figures

5-7). If the goal is to minimize the error in the total surface budget, then how does neglecting the flux divergence term contribute to the budget error? The authors should comment on this explicitly and note any caveats. It would also be a good idea to acknowledge this shortcoming and work towards evaluating these assumptions used when calculating storage terms.

> This matter is central to the evolving research campaign at the University of Tennessee. First, we see Eq. 2 as a gross approximation. The heat storage term of relevance to the surface heat budget must necessarily include heat stored in the biomass and in the layer of soil above the level of G determination. The accuracy of the measurement of G is another item that contributes to the resulting dilemma. The matter of heat storage in the air below the height of EC measurement was examined in the very early days of EC development and was ignored until studies over forests were started in the 1970s. These showed that the top priority was heat storage in the biomass, not in the air. Our use of Eq. 2 here is in recognition of what the ICOS community is doing, as an example of how the multiport system could be productively used. It should not be seen as an endorsement of the protocols adopted by ICOS.

> One of us (Oetting et al., 2024) has already examined the spatial uniformity issue, with results that support our present understanding that heterogeneity is likely to remain a problem with relying on eddy covariance methodologies. Better is to make relevant measurements as close to the surface as is possible and hence to minimize the consequences of the difficulties that arise.

> It is hard to separate flux divergence issues from the spatial heterogeneity (and topographic) problems that have already been addressed by this same group of workers (see various papers by O'Dell, Hicks, Eash and Oetting).

Height of vertical measurements: The authors note that the height of measurements was adjusted with the growth cycle during the two-month period. The authors should present a table or a stacked bar graph showing height adjustments to measurements as a function of time so that readers know how often the heights were adjusted and the range of heights that were used in this analysis. Furthermore, it is recommended and strongly encouraged that the storage term calculations be conducted for each set of measurement heights to highlight how the storage terms evolved during the growth cycle. The results in their current form mask the growth cycle, which is no doubt an In important contribution to $CO_2$ and $H_2O$ storage, photosynthetic response and evapotranspiration. I suspect that changing the height of measurements does not change the result too appreciably given profile linearity. However, I wonder how the slope of profile linearity changed during the growth cycle. These kinds of analyses would strengthen the paper and support the approach and instrument set-up presented in the paper.

- We have now included a table (Table 2) that presents the height adjustments of measurements throughout the growth cycle, along with corresponding changes in

the maximum and minimum storage terms. This table demonstrates how storage terms evolved with the crop's growth stages over the two-month study period. As the maize was in its early growth stage during the study, the canopy was not fully developed. Consequently, while the $CO_2$ storage did not show significant changes, there were substantial variations in the sensible and latent energy storage terms as the crop grew. These findings are now discussed in the revised manuscript, especially in relation to Table 2.

Shading in Figures 5-7: There is nothing in the text discussing the shading in these figures. I assume that the shading represents the variability within the two month period, but nothing is discussed about this shading. I do find it unlikely that there was almost no variability in CO2 during the day over the two month period, and the that the results are almost flat (hovering around 350 ppm). The authors need to add a discussion explaining the shading, and it is strongly encouraged to go back into the data to examine variability at each half-hour averaging interval examined during the diurnal cycle.

- Different colors of the figures correspond to different measurement heights. The shading of the plots represents bands of width +/- one standard error. The variability of CO2 was found to be higher at nighttime than in daytime. The highest variability was recorded within the canopy at height 1 (0.11 m) and height 2 (0.4- 1.4 m). For most of the daytime, the sub-canopy CO2 concentration remained at 350 ppm until sbout 0600 LT when the pooling of CO2 started.

Lines 302-303: The statement "...followed by a rapid decrease and negative storage until 2000 UTC" does not seem to support what is seen in Figure 7b. Sure, there was a sharp decline, but what followed was a fair amount of positive (accumulation) and negative (depletion) storage between this time period that does not lend support to this statement. The variability was large. Please revise messaging.

- Thank you for your observation. The figure 7 callout was wrong in the previous version and has been corrected in this revision.

Lines 295-297; 301-302: The authors claim that the sensible and latent fluxes exhibit similar characteristics. They do not.

- Yes, we agree with you, sensible and latent fluxes exhibit similar characteristics. We have updated the explanation in the revised manuscript.

Lines 288-290: The authors start out talking about morning and evening transitions, but only comment on the morning transition. This leads to a confusing interpretation if the reader views the statement made at the end of the sentence applicable to both morning and evening transitions when in fact it is not. Please comment on both and make the discussion on respective transitions clear.

- Thank you for your observation. We have now briefly described the morning and evening transition of $CO_2$-$H_2O$ fluxes, $CO_2$, latent and sensible heat storage in the revised manuscript.

Lines 295-297; 301-302: The authors claim that the sensible and latent fluxes exhibit similar characteristics. They do not. Lines 302-303: The statement "...followed by a rapid decrease and negative storage until 2000 UTC" does not seem to support what is seen in Figure 7b. Sure, there was a sharp decline, but what followed was a fair amount of positive (accumulation) and negative (depletion) storage between this time period that does not lend support to this statement. The variability was large. Please revise messaging.

- Thank you for your comment. We agree that the sensible and latent fluxes do not exhibit similar characteristics, and we have revised the manuscript to accurately reflect this. Yes we observed that there is positive (accumulation) and negative (depletion) storage between the indicated time period. We have revised the discussion to better reflect the observed fluctuations and variability in the data, as seen in Figure 7b.

Lines 304: I believe the authors can avoid the statement "presently unexplained" by studying Equation (2) and and examining Figure 6. This is where the slopes in the profile collapse and an inflection is observed (i.e., concentrations begin to decrease). So the time rate of change of concentrations using the discretized form in Equation (2) should help you form a hypothesis, especially since this is during a period where the stable boundary layer forms. It should also be noted that while the peak in Figure 7b stands out during 2000 UTC, it is not considerably larger than some of the peaks and troughs between 900 UTC and 2000 UTC that reach magnitudes close to 2.5 W/m2.

- Thank you for noticing this. We have removed the statement "presently unexplained". Yes, we agree that while the peak at 2000 UTC in Figure 7b stands out, it is not significantly larger than the other peaks and more variation can see between 0900 UTC and 2000 UTC.   It was recorded that laten heat storage magnitude reached close to 2 W m$^{-2}$ between 900 UTC and 2000 UTC when a stable boundary layer formed. This clarification has been incorporated into the revised manuscript.

Lines 303-305: Negligible? Visually I see statistically negative values during the night. The statistically negative latent storage (or depletion) follows a drop in water vapor concentration. The mechanisms for evapotranspiration are minimized and the uptake of moisture into the atmosphere near the surface is reversed as temperatures cool and condense at the surface.

- Thank you for your observation. Our use of the term "negligible" was intended to refer to the small positive storage of sensible energy during certain periods. We agree that, visually and statistically, negative latent storage (or depletion) occurs during the night, following a drop in water vapor concentration. At night temperature decreases, evapotranspiration minimized, and condensation occur. We have revised this section to clarify this point in the updated manuscript.

The authors sort of allude to this in lines 272-277, but fail to link these results when discussing Figure 7.

- Thank you for your observation. We agree that the link between the discussion in lines 272-277 and Figure 7 was not clearly made. In the revised manuscript, we have now explicitly connected the results discussed earlier with the interpretation of Figure 7. We elaborated on how the processes mentioned in lines 272-277, such as $CO_2$ and heat flux dynamics, directly relate to the patterns observed in Figure 7. This additional explanation provides a clearer narrative and helps readers better understand how these factors contribute to the overall results.

Lines 301-313: Discussion in this paragraph is filled with erroneous statements and broad remarks. Dive deeper into the analysis of this figure and make the interpretation clearer and stronger.

- Thank you for your comment. Actually, the current paper focused is just instrument and how this instrument can advance or update the science involved in the $CO_2$, $H_2O$ and fluxes and their storage terms measurement. Our next study is to provide the detail on the fluxes and storage terms. However, we have carefully reviewed and revised the discussion in the mentioned paragraph. In the revised version, we have provided a more in-depth analysis of all the figures and provide additional wind speed and precipitation figures to explain the underlying processes with stronger interaction of the factors. We have expanded the discussion with the support of relevant literature.

Conclusions: The key findings are not summarized nor are caveats and shortcomings of the approach laid out. The conclusions would also benefit from statements related to future work.

- Actually, the current paper has only focused on instrumentation. However, In the revised manuscript, we have included a more detailed summary of the key findings in the conclusion section. We highlighted the main outcomes of the study, including the effectiveness of the multiport system in measuring $CO_2$, heat, and water vapor profiles within and above the maize canopy. Additionally, we have addressed the limitations of the approach, such as potential errors in measurement due to sensor placement, power issue, external environmental factors like condensation etc. Furthermore, we have added statements on future work, including the need for further testing in different crop types and environments and its applications in agricultural, forestry and environmental research.

**Minor Comments:**

Line 49: Insert "the" between "improve" and "understanding"

- "The" has been added between improve understanding in the revised manuscript.

Line 56: can remove ", distributed"

- The word ", distributed" has been removed from the revised manuscript.

Lines 72-73: Comment on "loss of low- or high-frequency flux components": What does this mean exactly? Was a filter applied? Resolution limitation? Was it the technique used? Please elaborate and add a reference if possible

Line 73: Remove "etc."

- "etc." has been removed from revised manuscript

Lines 80-81: Perhaps instead say "Quantifying storage terms is challenging because measurements are required within and above the canopy". The current sentence is awkwardly phrased.

- The sentence has been rephrased according to the reviewer's comment.

Lines 107-108: remove "in their series of field experiments"

- "in their series of field experiments" has been removed from manuscript.

Line 110: Remove comma between "interest" and "within". There are a lot of other examples of misplaced commas.

- Commas has been removed between "interest" and "within". Furthermore, all extra commas have been removed.

Line 115: replaced "and" with "as" before "well".

"and" has been replaced with "as" in the revised manuscript.

Lines 120-121 and Lines 154-155: In the first set of lines the authors remark on cycling through all heights every minute while the second set of lines indicates every 30 seconds. Which is it?

- Thank you for pointing out the confusion. We apologize for not conveying this clearly. The system completes a full cycle of readings from all four heights in one minute. Specifically, it takes 7.5 seconds to take a reading from each height, and each height is measured twice (for a total of 15 seconds per height). Therefore, the entire process of reading from all four heights takes 60 seconds (15 seconds * 4 heights = 60 seconds/1 minute).

Line 195: change "variables" to "measurements"

- "variables" has been changed with "measurements" in revised manuscript.

Line 204: Add "in Figure 4" between "map" and "represents"

- "in Figure 4" has been added between "map" and "represents" in revised manuscript.

Lines 206-207: How did the authors account for the slope of terrain on sonic anemometer wind measurements since the plot indicates the circle is in 5-9% grade? Was the sonic anemometer orientated horizontally or was post processing done?

- The sonic anemometer was mounted and adjusted horizontally and precisely level. The measurement axis was parallel to gravity. Planner fit method as a post processing technique was applied for wind measurements using the ICOS methods

Line 207: How does the soil type factor into interpreting the results in this analysis? Soil type is important for moisture retention, capacity, and subsurface transport.

- Soil type is an important factor that impacts energy balance closure of the system by influencing physiochemical properties importantly soil temperature, moisture and other. Soil type strongly influences the storage and transfer of heat within the soil. In current study, we do consider soil heat storage, but this is of great interest to us, and we will consider it for our next canopy storage term study.

Line 226: Be clear about. Is it the separation distance between measurements? If so, then the thickness for the bottom measurement would be between the surface and the measurement height (0.11 m).

- Yes, $\Delta z_i$ is the vertical extent of the corresponding $i$th air layer (i.e. vertical segment). It represents the thickness of each layer (in our case thickness of each profile height) from the bottom of the 1$^{st}$ height (0.11m) towards the surface of the next measurement height.

Line 227: I'm guessing that "N" equals 4, but this not stated explicitly

- Yes, N is the number of sampling points as in our case N is equal to 4.

Line 237: "assumptions made elsewhere" should be backed up by citations.

- Citations (Galmiche and Hunt, 2002; Verma and Rosenberg, 1976) have been added in the revised manuscript.

Line 241: "with that aloft"? Or "with air aloft" or "with the overlying atmosphere"

Yes it "with the overlying atmosphere" and has been added in the revised manuscripts.

Line 245: can replace ", such that" with "as"

- ", such that" has been replaced with "as" in revised manuscript.

Paragraphs in lines 240-246 and 247-257 share a lot of redundant information

- The paragraphs in lines 240-246 and 247-257 have been rewritten and make the message clearer for readers.

Lines 299-300: "returning nearly to zero". Its not actually zero

- Yes, it is not zero, but it increases positively (accumulation) after 0700 LT, we updated in the revised manuscript.

Line 218: move "be" between "necessarily" and "site-specific" before "necessarily"

- "be" has been moved between "necessarily" and "site-specific" in revised manuscript.

Lines 252: Question on the sentence beginning with "Soil respiration". Are you referring to the entire diurnal trend, day portion or night portion? The sentence that preceded this one focused on the pattern at night, and as such, leads to confusion when interpreting the "Soil respiration" sentence.

- Yes, the sentence focuses on the night pattern but in the soil respiration sentence, we conclude in a single line by that the overall pattern of CO2 flux and storage is influenced by the diurnal change of soil respiration, photosynthesis, and temperature pattern.

This has been made clear in the revised manuscript.

Line 253: It is remarked that wind speeds decreased at night. Statistically, this may be true, but moderate to strong winds can develop at night which would effect the results and add to the variability observed and not discussed. At a minimum the authors should note that they confirmed the diurnal wind structure. They wouldn't need to show it, just make it clear that this was observed during the two-month survey period.

- Dear reviewer,We agree with your remarks on wind speed. Actually, this paper is focused on instrumentation, how these instruments help to better understand and examine the CO2 and H2O storage terms. How this instrument can advance science involve in modeling etc. Our 2nd studies focused on all these parameters,

  However, I have provided and explained the diurnal wind structure and precipitation for the two-month survey period.

The authors never discuss the time range that this study was conducted. Please indicate the months that this study took place before introducing Figures 5-7. This should be discussed in Section 3.

- The study was conducted from May to end of June 2023 and has been described in section 3 in the revised manuscripts.

Line 265: Replace ", we recorded" with "is"

- ", we recorded" has been replaced with "is" in revised manuscript.

Line 268: "height increased". The height of what increased?

- It means as the measurement height increases (from height toward height 4) $H_2O$ concentration decreased. It has been clear in the revised manuscript.

Line 271: The "profile appears to be stronger than" is not an appropriate description of the profile. I think what the authors meant to say is that the profile is steeper or that there is a more pronounced vertical gradient.

- The description in the revised manuscript has been updated with appropriate description according to author suggestions.

Line 271: Remove "cases" after H2O

- "cases" has been removed after H2O in the revised manuscript.

Line 272: Remove commas between "for H2O"

- Commas has been removed in the revised manuscript.

Figure 6 caption: You can omit "pattern" after "profile"

- "pattern" has been removed after "profile" in the revised manuscript

Line 286: Change "diurnal average" to "average diurnal".

- "diurnal average" has been changed to "average diurnal". In the revised manuscript.

Line 287: Change "higher values" to "larger magnitudes and more variation". Higher values could mean more positive. As such, I would instead go with the suggested change. Can also remove "as" between "nighttime" and "compared".

- We agree with the reviewer's suggestion. "higher values" has been changed with "larger magnitudes and more variation". Also "as" has been removed between "nighttime" and "compared" in the revised manuscript.

Line 295: Sensible heat energy storage was not always lower than latent heat storage. Also, references to "7b" and "7c" are incorrectly placed here and elsewhere in this section.

- We agree the sensible heat energy was not lower. I want to say that variation in the sensible energy storge was lower as compared to the latent energy storage. It is updated in the revised manuscript. We apologize for misplaced of 7b and 7c figure but placed correctly in the revised manuscript.

Lines 293-294: Comment on "During nighttime and morning, these processes were revered, leading to CO2 storage". During the morning transition the CO2 storage becomes negative (depletion) while during the night storage increases (accumulation). Make this clear instead of just saying "leading to CO2 storage".

- Thanks for the comments. It has been clearly explained in the revised manuscript that during nighttime and morning, these processes were revered, leading to CO2 storage. During the morning transition the CO2 storage becomes negative (depletion) while during the night storage increases (accumulation).

Missed opportunity connecting Figures 7a and 7b: The authors should dig into the inflected results between CO2 storage (7a) and latent energy storage (7b) that occur between 0630 UTC and 0900 UTC during the hours following sunrise. Note the importance of both CO2 and water to photosynthesis which activates at the time where the inflected behavior is observed.

- Soil microorganism activities and respiration from soil-pant surface increase over the sunrise as temperature increases, resulting in $CO_2$ release and accumulation near the soil surface and within the canopy which leads to observed $CO_2$ storage (Raza et al., 2022; Davidson and Janssens, 2006; Ryan and Law, 2005). More importantly, mostly at nighttime a stable boundary layer forms due to calm wind conditions which traps the $CO_2$. At sunrise due to temperature rise, the layer begins to weaken. The trapped $CO_2$ mixes with air, disperses, and decreases $CO_2$ storage on sunrise (Stull, 20012).
  The higher latent heat storage is due to the latent heat of vaporization, sunlight first hits the dews on the surface and initiates the evaporation which leads to greater latent heat flux, and water vapor stored in the atmosphere leads to higher latent heat storage (Jacobs and Nieveen, 1995).

Raza, T., Qadir, M.F., Khan, K.S., Eash, N.S., Yousuf, M., Chatterjee, S., Manzoor, R., ur Rehman, S. and Oetting, J.N., 2023. Unrevealing the potential of microbes in decomposition of organic matter and release of carbon in the ecosystem. *Journal of Environmental Management*, *344*, p.118529.

Davidson, E. A., & Janssens, I. A. (2006). Temperature sensitivity of soil carbon decomposition and feedbacks to climate change. *Nature*, 440(7081), 165-173. doi:10.1038/nature04514.

Ryan, M. G., & Law, B. E. (2005). Interpreting, measuring, and modeling soil respiration. *Biogeochemistry*, 73(1), 3-27. doi:10.1007/s10533-004-5167-7

Jacobs, A.F. and Nieveen, J.P., 1995. Formation of dew and the drying process withincrop canopies. *Meteorological Applications*, *2*(3), pp.249-256.

Stull, R.B., 2012. *An introduction to boundary layer meteorology* (Vol. 13). Springer Science & Business Media.

Line 298: change "After that, this" to "Afterwards the"

- "After that, this" has been changed to "Afterwards the" in the revised manuscript.

Lines 299-300: "returning nearly to zero". Its not actually zero

- Thank you for the observation. We agree with the reviewer that the values do not actually return to zero, but instead revert to positive (accumulation) values. We have made the necessary clarification and updated the wording accordingly in the revised manuscript.

Line 305: Insert "afternoon into" between "the" and "late nighttime

- "afternoon into" has been inserted between "the" and "late nighttime" in the revised manuscript.

Figure 7: There is not a comment about how the data is being processed to form the plots. Is the data vertically integrated? I'm guessing not given the units. If so, what height are the authors choosing or is vertical averaging done? Y-axis label in Figure 7b has a parenthesis that shouldn't be there. Please place the a-c labeling in a corner and not over the plots.

- The data processing involved averaging the raw data into 15-minute intervals. The data was analyzed using EddyPro 4.1 software, which processes the raw data by applying block averaging and filtering out outliers based on Reynolds decomposition. Additionally, statistical analysis was performed in R Studio to generate the plots that depict the variation. This approach ensures accurate representation and analysis of the storage terms. Yes, the data is vertically integrated in this instance. A parenthesis in fig 7b has been removed and labeled has been placed in the corner of the graph in the revised manuscript.

Line 313: Change from "the accurate estimation of flux" to "the accurate estimation of fluxes"

- "the accurate estimation of flux" has been changed to "the accurate estimation of fluxes" in the revised manuscript.

Figure 7a: How do you explain the large variation between negative and positive CO2 at night and virtually no storage/no variability during the day?

- During the morning to onward, transition of the $CO_2$ storage becomes negative (depletion) while during the night storage increases (accumulation). Soil microorganism activities and respiration from soil-pant surface increase over the sunrise as temperature increases, resulting in $CO_2$ release and accumulation near the soil surface and within the canopy which leads to observed $CO_2$ storage (Raza et al., 2022; Davidson and Janssens, 2006; Ryan and Law, 2005). More importantly, mostly at nighttime a stable boundary layer forms due to calm wind conditions which traps the $CO_2$. At sunrise due to temperature rise, the layer begins to weaken. The trapped $CO_2$ mixes with air, disperses, and decreases $CO_2$ storage on sunrise (Stull, 20012).

This section has been added to the revised manuscript.

Figure quality needs general improvement. Larger fonts, particularly in Figures 5-7 is needed.

- Thank you for the suggestion. The figure quality has been improved in the revised manuscript, and larger fonts have been applied, particularly in Figures 5-7, to enhance readability.

---

## Author Comment (AC2)

**Date:** Tuesday, November 12, 2024

To,

Russel Dickerson
Handling Editor
**Atmospheric Measurement Techniques**

Sub.:- Submission revision RC2 of research paper to the atmospheric measurement techniques

Dear Sir,

Greetings of the day, I hope you are doing well! With reference to the above-cited subject, we are resubmitting herewith the revised version of research paper on **"A simplified system to quantify carbon dioxide, water vapor and heat within a maize canopy (egusphere-2024-2041)"** to your esteemed journal **"atmospheric measurement techniques".** Authors are grateful to the reviewer for a careful and helpful analysis of our manuscript. Undoubtedly, due to the Reviewer`s comments, the manuscript has been significantly improved. All the reviewer`s comments are reasonable, and we have corrected the MS in accordance with the comments and suggestions

Thank you for receiving our manuscript and considering it for further processing. Authors appreciate your valuable time and look forward to your response.

Sincerely yours,
Corresponding author
Taqi Raza
taqiraza85@gmail.com, traza@vols.utk.edu
Department of Biosystems Engineering & Soil Science
University of Tennessee, Knoxville USA

**Major comments**

the presentation of Fig. 2, especially regarding the transit/sampling time, needs further clarification. The first question is: should we expect the same pattern for intake 2 as shown for intake 1? I do see the first few seconds data of intake 1 should be discarded while the remaining data are stable and good for analysis. But where are the stable, equilibrium regions for intakes #2 and #4? I presume that a similar stable region and the same amount of data points are desired for each height/intake.

✓ Yes, not only 1 and 2 but all intake tubes have the same sampling time. After careful consideration, we decided to remove Table 1 (IRGA output for the multiport air sampling system, for CO2 conc. (430 ppm) fed through the sampling lines at ambient pressure) and Figure 2 (The time-dependent relationship between the infrared gas analyzer (IRGA) in the multiport air sampling apparatus for a gas concentration of 430 ppm $CO_2$ flowing at <1 L/min) from the manuscript, as it raised additional questions and complexities not central to the primary objectives of our study. The initial purpose of this figure was to demonstrate that the instrument was accurately measuring a known $CO_2$ concentration (430 ppm) using different intake tubes, which was a verification step supporting the field trial described in the text that follows. In the revised manuscript, we have clarified that both $N_2$ (0 ppm $CO_2$) and $CO_2$ (430 ppm) were used to test the stability of the system and confirm that the analyzer and switching mechanism were functioning correctly. We believe that the removal of this figure enhances the clarity and focus of the manuscript.

The second question is: at line 181, the authors mentioned that the 4.9 s delay time was improved to 3.2s but did not tell how this was achieved. Did the authors either increase the flow rate or change the tube length? Also, why not show a figure with 3.2 seconds delay since it's the most recent and reasonable result? Using 4.9 seconds delay is confusing here.

✓ Thank you for bringing this to our attention. We made a mistake. In fact, actual timing for measurement was 3.1 seconds and 4.4 seconds were ignored during the $CO_2$ reading by the analyzer. This has been briefly explained in the revised manuscript.

2) This concern is about section 3.1 Experimental Site. First, the authors started this section with introducing their field instruments and setup, then move to a subsection talking about the basics about the field site. Would it be logically better to introduce the experimental site first, then describe the field measurement setup?

✓ This is the valid question of the reviewer, and we agree that logically experiment site should first and then the field measurement setup. In the revised manuscript we have incorporated the reviewer's suggestion

Second, should subsection 3.2 belong to Methodology? It's odd to put it under the section Field Measurement Setup.

✓ Thank you for bringing this to our attention. We have moved subsection 3.2 under the result section.

3) Another concern is about section 4. First, starting at line 247, this paragraph delivers almost the same message as the paragraph above and adds nothing new. Second, starting at line 295, Fig.7b and 7c are mixed up in the figure and in the text, making it hard for readers to follow. The rest of this section lacks further discussion on the results, such as the advantages/limitations of this system compared to other similar studies/systems, a more comprehensive description on what aspects this system can be useful for in future micromet studies.

✓ Thank you for your feedback. We have addressed the concerns you raised in the revised manuscript. We have removed the reputation you mentioned in the revised manuscript. We have also fixed the figure 7 a, b & c in the revised manuscript. We have given a complete paragraph describing how this study can be useful for future micromet studies at the end of the result section. Similarly, the limitation of the study has been given in the conclusion and summary. See the revised manuscript.

**Minor comments are also listed below:**

Highlights: the second and third bullet points convey the same meaning: neglecting the storage terms leads to inaccuracies (point 2) while considering it leads to an improved accuracy (point 3). In addition, how this consideration of storage fluxes improved the accuracy of energy closure was not shown in the study. Full budget calculations with/without the storage terms are needed to prove that.

✓ Yes, we agree with that highlight 2 & 3 conveying same message. We have removed point 2 and added new highlights. We agree with the reviewer that how this consideration of storage fluxes improved the accuracy of energy closure was not shown in the study. Actually, this paper focuses on the development of the system to measure the storage terms for considering them in the energy balance closure. How this instrument can advance science involve in modeling etc. Our 2$^{nd}$ study focused on calculations of full budget with/without the storage terms.

2) Line 53, the previous two references (Lamas Galdo et al., 2021 and Wang et al., 2023) are very recent, can you find a more recent one to replace this 2004 paper?

✓ Recent references have been added in the revised manuscript (e.g., Hoeltgebaum and Nelson, 2023).

3) Line 56, same as above. Replace this one with a more recent paper if possible.

✓ Recent reference (Varmaghani et al., 2016) have been added in the revised manuscript

4) Line 67, is there a newer paper discussing how many sites are measuring storage? This number must have increased since 2006.

✓ We have updated the number of sites in the revised manuscript and also provided recent references (Fluxnet; Pastorello et al., 2020).

Line 108, a period is missing before 'These'.

✓ Thak you for your attention. A period has been added before "these" in the revised manuscript.

Line 115, the first 'and' changes to 'as'.

✓ Thak you for your attention. "as" has been replaced with "and" in the revised manuscript.

Line 115, 'To minimize consequences of individual sensor offsets". My comment: Actually you avoided the consequence of inconsistency in sensor offsets because you were using a single sensor rather than multiple, so I think you could just say "avoid" instead of "minimize" because it was no longer an issue.

✓ Thank you for your comments. "minimize" has been replaced with "avoid" in the revised manuscript.

Line 162, '3.2 seconds were ignored' changes to 'The first 3.2 seconds of data were discarded, and the remaining ...'

✓ Thank you for bringing this to our attention. We made a mistake.  In fact, actual timing for measurement was 3.1 seconds and 4.4 seconds were ignored during the $CO_2$ reading by the analyzer. This has been briefly explained in the revised manuscript. These time was monitored during the laboratory evaluation of the system for our understanding.

9) Line 162, maybe I missed that but I don't think you have mentioned the frequency of your data so far. Is it 10Hz, 20Hz or else? Since you start to talk about data here, readers may wonder how many data points are there in 4.3 seconds.

✓ The frequency of data was 10Hz which is mentioned in 2.1 Apparatus design and operation. After careful consideration, we decided to remove Table 1 (IRGA output for the multiport air sampling system, for CO2 conc. (430 ppm) fed through the sampling lines at ambient pressure) and Figure 2 (The time-dependent relationship between the infrared gas analyzer (IRGA) in the multiport air sampling apparatus for a gas concentration of 430 ppm $CO_2$ flowing at <1 L/min) from the manuscript, as it raised additional questions and complexities not central to the primary objectives of our study. The initial purpose of this figure was to demonstrate that the instrument was accurately measuring a known $CO_2$ concentration (430 ppm) using different intake tubes, which was a verification step supporting the field trial described in the text that follows. In the revised manuscript, we have clarified that both $N_2$ (0 ppm $CO_2$) and $CO_2$ (430 ppm) were used to test the stability of the system and confirm that the analyzer and switching mechanism were functioning

correctly. We believe that the removal of this figure enhances the clarity and focus of the manuscript.

10) Line 164, I don't get this 1.8 seconds. You just said 3.2 seconds is the delay time but now you have another delay time. Is the 3.2 seconds delay for the entire switching system and the 1.8 s delay for the IRGA separately? If you did not use this 1.8s delay time for your flux data calculation, I think it would be better to not mention it to avoid confusion.

✓ 1.8 seconds mean system takes 1.8 second to produce steady or continuous reading of $CO_2$, and $H_2O$. After 1.8 seconds there is no gap (missing data points) recorded. Yes, we did not use this data recorded during this delay period for flux calculation. The 1.8 second is out of 3.2 seconds. The timing was just for our record to check the response time.

11) I don't see any results about the environmental data such as precipitation, temperature, wind, etc. Such data could provide useful information or at least context about this specific location.

✓ A new table 1 is added where precipitation, temperature and storage terms data are provided along the cop height and growth stage. For wind speed see the figure 4. Both table 1 and figure 4 are added into the revised manuscript.

12) Line 211, 'A)' for what?

✓ Thank you for pointing out the mistake. "A)" has been deleted in the revised manuscript.

13) Line 213, font size changed.

✓ The front size has been fixed in the revised manuscript.

14) Line 260, times of sunrise and sunset are NOT shown.

✓ We have deleted "times of sunrise and sunset are shown" from the description of figure 5 and described the time of sunrise and sunset in section 3 experimental site in the revised manuscript.

15) Line 262, '15-minute'. My comment: this is the only place you mentioned the average time for flux calculation. You should do this earlier in the method section.

✓ Thank you for bringing your attention. "constructed from 15-minute" has been deleted as It was already described in the methodology section (2.1 Apparatus design and operation).

16) Fig. 5 and Fig. 6: Need to provide a separate table/graph showing how the heights of the intakes were adjusted during the two-month study period (at least in the supplementary materials). Moreover, the sampling heights in these figures are not consistent with what the authors described earlier. For instance, in the figures, the maximum height for the

second level is 1.4 m (0.5h), which means max canopy height h is 2.8 m. Then the max height for the third level (1+h) should be 3.8 m and the max for the top level (2+h) should be 4.8 m. However, in either figure the maximum heights are only 3 m for the third level and 4.4 m for the fourth level. This inconsistency needs explanation.

✓ The instrument evaluation study was conducted for six weeks from May to end of June 2023 (for your clearance, results are provided for only 6 weeks measurement). The maximum height for H1, H2, H3, and H4 was 0.11m, 1.27m, 3m and 4.36m respectively. In the revised manuscript the measurements along the height and growth stage are provided in table 1.

17) Line 303, there were both positive and negative storage before 2000LT.

✓ Latent heat storage (Fig. 7c) fluctuated about zero for most of the daylight hours, after exhibiting a major positive excursion (> 4 W m$^{-2}$) during the few hours after sunrise. After about 2100 LT, similar fluctuations occurred until sunrise, with an average of about –0.5 W m$^{-2}$. We have explained it clearly in the revised manuscript.

18) Fig.7: figure letters a-c need to be placed at the same corner.

✓ We apologize for misplaced letter a-c in fig 7 but placed correctly in the same corner in the revised manuscript

---

## Author Response (AR1)

**Date:** Sunday, April 20, 2025

To,

Dr. Russell Dickerson
Handling Editor
**Atmospheric Measurement Techniques**

Subject   Submission revision RC1-3 of research paper **egusphere-2024-2041** to Atmospheric Measurement Techniques

Dear Sir,

We are resubmitting herewith a revised version of our research paper on **"A simplified system to quantify storage of carbon dioxide, water vapor and heat within a maize canopy (egusphere-2024-2041)".** We are grateful to the reviewers for their careful and helpful analysis of our manuscript. Based on reviewers' comments, the manuscript has been substantially improved.

The main point of our submission is that a new and very reliable system has been developed for measuring gradients of water vapor and carbon dioxide within the surface roughness layer. To substantiate the utility of our development, we have tested it as a part of a larger field campaign. Detailed analysis of the field experiment's observations will follow. For the present, all we want to do is provide information to describe how we are obtaining data and to alert others who are making measurements in similar demanding circumstances. The reviewer's complaints that we are not presenting the results of the host field program are therefore difficult to deal with and we have consequently removed all text that could confuse this matter.

Thank you for receiving our manuscript and considering it for further processing. The authors appreciate your valuable time and look forward to your response.

Sincerely yours,

Corresponding author

Taqi Raza

taqiraza85@gmail.com, traza@vols.utk.edu

Department of Biosystems Engineering & Soil Science

University of Tennessee, Knoxville USA

**Reviewer 1**

First, we must emphasize our regret that we failed to explain the reason for devising the sampling system now described. This has now been rectified, and we hope that some of the Reviewer's misgivings will be assuaged.

In brief, this development is a part of a decade-long program of the University of Tennessee (UT) to refine understanding of maize agriculture in locations outside the conventional realm of perfect sites. The focus is on small farms in complex terrain, such as dominating most of Africa and much of eastern North America. Experience has resulted in a need for detailed $CO_2$ and water vapor gradients in the region below where conventional micrometeorology applies. The system must be sensitive and capable of reliable operation at sites distant from technical support. This background material has now been emphasized, with modifications mainly appearing in the Introductory discussion.

This new sampling mechanism will constitute a major part of all of the research studies to follow as part of this UT research program. It is our desire to prepare for these future studies by bringing this development to the fore, as well as to make the external agricultural and micrometeorological communities aware of our progress.

It is not our intent to present results from the intensive field campaign of 2023 that made first field use of our sampler, other than to demonstrate how the new device can contribute to the research program in a constructive, reliable and cost-effective manner. Detailed findings of the 2023 campaign will be presented elsewhere in due course.

**Major Comments**

Comments: Figure 2: In the caption, the authors note that the vertical lines represent stable periods where measurements are suitable for recording. The vertical lines and how they are placed in the figure do not seem to support the text in lines 148-151. The text suggests that feeding in a known amount of CO2 (430 ppm) and looking at stability of these measurements define the periods of acceptable data. Based on Figure 1, intake tubes 1 and 2 (3 and 4) are coupled with a smooth transition from one intake tube to the other, while rapid increases and decreases occur as the measurement sequence switches from 2 to 3 and 4 to 1. If useful data is defined by obtaining the ambient CO2 for this test example, then the stability regions in the plot should be represented by time periods between the

vertical solid and dashed lines in the annotated Figure 2 plot shared below. If this is not the case, then the authors need to improve the explanation of this plot, because it would seem undesirable to isolate and use the measurements when CO2 is 0 ppm as suggested in Figure 2 with the second and fourth vertical black lines. Lastly, why do measurement heights for tubes 2 and 4 record less data with respect to time compared to tubes 1 and 3 when measuring 430 ppm of CO2? I would think that achieving an approximately equal number of data points for each measurement height is preferred. Please address this.

✓ After careful consideration, we decided to remove Table 1 (IRGA output for the multiport air sampling system, for CO2 conc. (430 ppm) fed through the sampling lines at ambient pressure) and Figure 2 (The time-dependent relationship between the infrared gas analyzer (IRGA) in the multiport air sampling apparatus for a gas concentration of 430 ppm $CO_2$ flowing at <1 L/min) from the manuscript, as it raised additional questions and complexities not central to the primary objectives of our study. The initial purpose of this figure was to demonstrate that the instrument was accurately measuring a known $CO_2$ concentration (430 ppm) using different intake tubes, which was a verification step supporting the field trial described in the text that follows. In the revised manuscript, we have clarified that both $N_2$ (0 ppm $CO_2$) and $CO_2$ (430 ppm) were used to test the stability of the system and confirm that the analyzer and switching mechanism were functioning correctly. We believe that the removal of this figure enhances the clarity and focus of the manuscript.

Comments: Lines 179-181: If the authors were able to improve the set-up to measure at 3.2 seconds compared to 4.9 seconds, then the authors should show the plot with improved timing. The authors can show both figures with a panel dedicated to each that shows contrast. This is strongly encouraged.

✓ Thank you for bringing this to our attention. We made a mistake. In fact, actual timing for measurement was 3.1 seconds and 4.4 seconds were ignored during the $CO_2$ reading by the analyzer. This has been briefly explained in the revised manuscript.

Equation 2: The form of this equation uses two assumptions defined in Montagnani et. al. 2018: a homogeneous footprint and that the flux divergence can be ignored. While I have more confidence in the first assumption given that the survey region is a large maize field, I have concerns about neglecting the flux divergence term, especially during transitional periods (i.e., sunrise/sunset) where these calculations seem very important (e.g., Figures 5-7). If the goal is to minimize the error in the total surface budget, then how does neglecting the flux divergence term contribute to the budget error? The authors should comment on this explicitly and note any caveats. It would also be a good idea to acknowledge this shortcoming and work towards evaluating these assumptions used when calculating storage terms.

✓ This matter is central to the evolving research campaign at the University of Tennessee. First, we see Eq. 2 as a gross approximation. The heat storage term of relevance to the surface heat budget must necessarily include heat stored in the biomass and in the layer of soil above the level of G determination. The accuracy of the measurement of G is another item that contributes to the resulting dilemma. The

matter of heat storage in the air below the height of EC measurement was examined in the very early days of EC development and was ignored until studies over forests were started in the 1970s. These showed that the top priority was heat storage in the biomass, not in the air. Our use of Eq. 2 here is in recognition of what the ICOS community is doing, as an example of how the multiport system could be productively used. It should not be seen as an endorsement of the protocols adopted by ICOS.

One of us (Oetting et al., 2024) has already examined the spatial uniformity issue, with results that support our present understanding that heterogeneity is likely to remain a problem with relying on eddy covariance methodologies.  Better is to make relevant measurements as close to the surface as is possible and hence to minimize the consequences of the difficulties that arise with EC deployment at greater heights..

It is hard to separate flux divergence issues from the spatial heterogeneity (and topographic) problems that have already been addressed by this same group of workers (see various papers by O'Dell, Hicks, Eash and Oetting).

Height of vertical measurements: The authors note that the height of measurements was adjusted with the growth cycle during the two-month period. The authors should present a table or a stacked bar graph showing height adjustments to measurements as a function of time so that readers know how often the heights were adjusted and the range of heights that were used in this analysis. Furthermore, it is recommended and strongly encouraged that the storage term calculations be conducted for each set of measurement heights to highlight how the storage terms evolved during the growth cycle. The results in their current form mask the growth cycle, which is no doubt an In important contribution to CO2 and H2O storage, photosynthetic response and evapotranspiration. I suspect that changing the height of measurements does not change the result too appreciably given profile linearity. However, I wonder how the slope of profile linearity changed during the growth cycle. These kinds of analyses would strengthen the paper and support the approach and instrument set-up presented in the paper.

✓ We have now included a table (Table 2) that presents the height adjustments of measurements throughout the growth cycle, along with corresponding changes in the maximum and minimum storage terms. This table demonstrates how storage terms evolved with the crop's growth stages over the two-month study period. As the maize was in its early growth stage during the study, the canopy was not fully developed. Consequently, while the $CO_2$ storage did not show significant changes, there were substantial variations in the sensible and latent energy storage terms as the crop grew. These findings are now discussed in the revised manuscript, especially in relation to Table 2.

Shading in Figures 5-7: There is nothing in the text discussing the shading in these figures. I assume that the shading represents the variability within the two month period, but nothing is discussed about this shading. I do find it unlikely that there was almost no variability in CO2 during the day over the two month period, and the that the results are

almost flat (hovering around 350 ppm). The authors need to add a discussion explaining the shading, and it is strongly encouraged to go back into the data to examine variability at each half-hour averaging interval examined during the diurnal cycle.

✓ The figure has been revised to present results in a more familiar manner, with averages and standard error bounds shown.  As expected, the variability of $CO_2$ was found to be higher at nighttime than in daytime. The highest variability was recorded within the canopy at height 1 (0.11 m) and height 2 (0.4- 1.4 m). For most of the daytime, the sub-canopy $CO_2$ concentration remained at 350 ppm until about 0600 LT when the pooling of $CO_2$ started.   Details like these are provided by the new sampling system and are provided here as evidence of the system's field utility.

We have eliminated much of the discussion that would be better presented separately, as a component of a research paper describing the 2023 field experiment, its data analysis and its conclusions. Nevertheless, we offer responses to the Reviewer's comments as follow.

Lines 302-303: The statement "…followed by a rapid decrease and negative storage until 2000 UTC" does not seem to support what is seen in Figure 7b. Sure, there was a sharp decline, but what followed was a fair amount of positive (accumulation) and negative (depletion) storage between this time period that does not lend support to this statement. The variability was large. Please revise messaging.

✓ The figure 7 callout was wrong in the previous version and has been corrected in this revision. We appreciate that we need to be careful about claiming research results when the main goal is to promote the field utility of the new sampling system we have developed.

Lines 295-297; 301-302: The authors claim that the sensible and latent fluxes exhibit similar characteristics. They do not.

✓ Yes, we agree with you, sensible and latent fluxes differ substantially. This discussion is not relevant to the main purpose of this paper and has been taken out.

Lines 288-290: The authors start out talking about morning and evening transitions, but only comment on the morning transition. This leads to a confusing interpretation if the reader views the statement made at the end of the sentence applicable to both morning and evening transitions when in fact it is not. Please comment on both and make the discussion on respective transitions clear.

✓ This matter is peripheral to the main purpose of our paper and we have made corrections accordingly.

Lines 295-297; 301-302: The authors claim that the sensible and latent fluxes exhibit similar characteristics. They do not. Lines 302-303: The statement "…followed by a rapid decrease and negative storage until 2000 UTC" does not seem to support what is seen in Figure 7b. Sure, there was a sharp decline, but what followed was a fair amount of positive

(accumulation) and negative (depletion) storage between this time period that does not lend support to this statement. The variability was large. Please revise messaging.

✓ We agree that the sensible and latent fluxes do not exhibit similar characteristics, and we have revised the manuscript to accurately reflect this.

Lines 304: I believe the authors can avoid the statement "presently unexplained" by studying Equation (2) and and examining Figure 6. This is where the slopes in the profile collapse and an inflection is observed (i.e., concentrations begin to decrease). So the time rate of change of concentrations using the discretized form in Equation (2) should help you form a hypothesis, especially since this is during a period where the stable boundary layer forms. It should also be noted that while the peak in Figure 7b stands out during 2000 UTC, it is not considerably larger than some of the peaks and troughs between 900 UTC and 2000 UTC that reach magnitudes close to 2.5 W/m2.

✓ Thank you for noticing this. We have removed the statement "presently unexplained". Yes, we agree that while the peak at 2000 UTC in Figure 7b (now Fig. 6b) stands out, it has the appearance of an anomaly yet to be explained. Our text now addresses this.

Lines 303-305: Negligible? Visually I see statistically negative values during the night. The statistically negative latent storage (or depletion) follows a drop in water vapor concentration. The mechanisms for evapotranspiration are minimized and the uptake of moisture into the atmosphere near the surface is reversed as temperatures cool and condense at the surface.

• Our use of the term "negligible" was intended to refer to the small positive storage of sensible energy during certain periods. We agree that, visually and statistically, negative latent storage (or depletion) occurs during the night, following a drop in water vapor concentration. At night temperature decreases, evapotranspiration minimized, and condensation occur. We have revised this section to clarify this point in the updated manuscript.

The authors sort of allude to this in lines 272-277, but fail to link these results when discussing Figure 7.

✓ In the revised manuscript, we have now explicitly connected the results discussed earlier with the interpretation of Figure 7 (6).

Lines 301-313: Discussion in this paragraph is filled with erroneous statements and broad remarks. Dive deeper into the analysis of this figure and make the interpretation clearer and stronger.

✓ Thank you for your comment. Actually, the current paper focused is just on instrumentation and how this instrument can advance or update the science involved in the CO2 and H2O fluxes and their atmospheric storage terms. However, we have carefully reviewed and revised the discussion so as to make sure the reader does not expect a final report on a major field campaign when our intent is no more than to describe and promote the instrumentation with which we obtained the data.

Conclusions: The key findings are not summarized nor are caveats and shortcomings of the approach laid out. The conclusions would also benefit from statements related to future work.

✓ Actually, the current paper has only focused on instrumentation. We have added mention of proposed future work.

**Minor Comments:**

✓ Our revision has been with awareness of all of the following "minor comments." In many cases, the discussion of concern has now been eliminated, as being tangential to the main purpose of our presentation.

Line 49: Insert "the" between "improve" and "understanding"

✓ "The" has been added between improve understanding in the revised manuscript.

Line 56: can remove ", distributed"

✓ The word ", distributed" has been removed from the revised manuscript.

Lines 72-73: Comment on "loss of low- or high-frequency flux components": What does this mean exactly? Was a filter applied? Resolution limitation? Was it the technique used? Please elaborate and add a reference if possible

✓ No filtering was used. Our comment relates to the need to rely on high frequency observations and long sampling times. Whereas many rely on 10 Hz data collection, we used 5 Hz. We examined the need for a correction and found none. The low frequency issue relates to the selection of run times, which should be long enough to reflect the entire breadth of the cospectrum. We used 15 min run times whereas many workers prefer longer. We found no reason to change.

Massman, W. and Lee, X.: Eddy covariance flux corrections and uncertainties in long-flux 466 studies of carbon and energy exchanges, Agric. For. Meteorol., 113(1-4), 121-144, 467 https://doi.org/10.1016/S0168-1923(02)00105-3, 2002.

Line 73: Remove "etc."

✓ "etc." has been removed from revised manuscript

Lines 80-81: Perhaps instead say "Quantifying storage terms is challenging because measurements are required within and above the canopy". The current sentence is awkwardly phrased.

✓ The sentence has been rephrased according to the reviewer's comment.

Lines 107-108: remove "in their series of field experiments"

✓ "in their series of field experiments" has been removed from manuscript.

Line 110: Remove comma between "interest" and "within". There are a lot of other examples of misplaced commas.

✓ Commas has been removed between "interest" and "within". Furthermore, all extra commas have been removed.

Line 115: replaced "and" with "as" before "well".

✓ "and" has been replaced with "as" in the revised manuscript.

Lines 120-121 and Lines 154-155: In the first set of lines the authors remark on cycling through all heights every minute while the second set of lines indicates every 30 seconds. Which is it?

✓ Thank you for pointing out the confusion. We apologize for not conveying this clearly. The system completes a full cycle of readings from all four heights in one minute. Specifically, it takes 7.5 seconds to take a reading from each height, and each height is measured twice (for a total of 15 seconds per height). Therefore, the entire process of reading from all four heights takes 60 seconds (15 seconds * 4 heights = 60 seconds/1 minute).

Line 195: change "variables" to "measurements"

✓ "variables" has been changed with "measurements" in revised manuscript.

Line 204: Add "in Figure 4" between "map" and "represents"

✓ "in Figure 4" has been added between "map" and "represents" in revised manuscript.

Lines 206-207: How did the authors account for the slope of terrain on sonic anemometer wind measurements since the plot indicates the circle is in 5-9% grade? Was the sonic anemometer orientated horizontally or was post processing done?

✓ The sonic anemometer was mounted and adjusted horizontally and precisely level. The measurement axis was according to gravity. Planar fit method as a post processing technique was applied for wind measurements using the ICOS methods.

Line 207: How does the soil type factor into interpreting the results in this analysis? Soil type is important for moisture retention, capacity, and subsurface transport.

✓ Soil type is an important factor that impacts energy balance closure of the system by influencing physiochemical properties importantly soil temperature, moisture and other. Soil type strongly influences the storage and transfer of heat within the soil. In current study, we do consider soil heat storage, but this is of great interest to us, and we will consider it for our next canopy storage term study.

Line 226: Be clear about. Is it the separation distance between measurements? If so, then the thickness for the bottom measurement would be between the surface and the measurement height (0.11 m).

✓ Yes, $\Delta z_i$ is the vertical extent of the corresponding $i$th air layer (i.e. vertical segment). It represents the thickness of each layer (in our case thickness of each profile height) from the bottom of the 1st height (0.11m) towards the surface of the next measurement height.

Line 227: I'm guessing that "N" equals 4, but this not stated explicitly

✓ Yes, N is the number of sampling points as in our case N is equal to 4.

Line 237: "assumptions made elsewhere" should be backed up by citations.

✓ Citations (Galmiche and Hunt, 2002; Verma and Rosenberg, 1976) have been added in the revised manuscript.

Line 241: "with that aloft"? Or "with air aloft" or "with the overlying atmosphere"

✓ Yes it "with the overlying atmosphere" and has been added in the revised manuscripts.

Line 245: can replace ", such that" with "as"

✓ ", such that" has been replaced with "as" in revised manuscript.

Paragraphs in lines 240-246 and 247-257 share a lot of redundant information

✓ The paragraphs in lines 240-246 and 247-257 have been rewritten and make the message clearer for readers.

Lines 299-300: "returning nearly to zero". Its not actually zero

✓ Yes, it is not zero, but it increases positively (accumulation) after 0700 LT, we updated in the revised manuscript.

Line 218: move "be" between "necessarily" and "site-specific" before "necessarily"

✓ "be" has been moved between "necessarily" and "site-specific" in revised manuscript.

Lines 252: Question on the sentence beginning with "Soil respiration". Are you referring to the entire diurnal trend, day portion or night portion? The sentence that preceded this one focused on the pattern at night, and as such, leads to confusion when interpreting the "Soil respiration" sentence.

✓ Yes, the sentence focuses on the night pattern but in the soil respiration sentence, we conclude in a single line by that the overall pattern of CO2 flux and storage is influenced by the diurnal change of soil respiration, photosynthesis, and temperature pattern. This has been made clear in the revised manuscript.

Line 253: It is remarked that wind speeds decreased at night. Statistically, this may be true, but moderate to strong winds can develop at night which would effect the results and add to the variability observed and not discussed. At a minimum the authors should note that they confirmed the diurnal wind structure. They wouldn't need to show it, just make it clear that this was observed during the two-month survey period.

✓ We agree with your remarks on wind speed. Actually, this paper is focused on instrumentation, how these instruments help to better understand and examine the CO2 and H2O storage terms. How this instrument can advance science involve in modeling etc. Our 2nd study focused on all these parameters.

However, we have provided and explained the diurnal wind structure and precipitation for the two-month survey period.

The authors never discuss the time range that this study was conducted. Please indicate the months that this study took place before introducing Figures 5-7. This should be discussed in Section 3.

✓ The study was conducted from May to end of June 2023 and has been described in section 3 in the revised manuscript.

Line 265: Replace ", we recorded" with "is"

✓ ", we recorded" has been replaced with "is" in revised manuscript.

Line 268: "height increased". The height of what increased?

✓ It means as the measurement height increases (from height 1 to height 4) $H_2O$ concentration decreased. This has been clarified in the revised manuscript.

Line 271: The "profile appears to be stronger than" is not an appropriate description of the profile. I think what the authors meant to say is that the profile is steeper or that there is a more pronounced vertical gradient.

✓ The description in the revised manuscript has been updated. Line 271: Remove "cases" after H2O

✓ "cases" has been removed after H2O in the revised manuscript.

Line 272: Remove commas between "for H2O"

✓ Commas has been removed in the revised manuscript.

Figure 6 caption: You can omit "pattern" after "profile"

✓ "pattern" has been removed after "profile" in the revised manuscript

Line 286: Change "diurnal average" to "average diurnal".

✓ "diurnal average" has been changed to "average diurnal". In the revised manuscript.

Line 287: Change "higher values" to "larger magnitudes and more variation". Higher values could mean more positive. As such, I would instead go with the suggested change. Can also remove "as" between "nighttime" and "compared".

✓ We agree with the reviewer's suggestion. "higher values" has been changed with "larger magnitudes and more variation". Also "as" has been removed between "nighttime" and "compared" in the revised manuscript.

Line 295: Sensible heat energy storage was not always lower than latent heat storage. Also, references to "7b" and "7c" are incorrectly placed here and elsewhere in this section.

✓ We agree the sensible heat energy was not lower. I want to say that variation in the sensible energy storge was lower as compared to the latent energy storage. It is updated in the revised manuscript. We apologize for misplaced of 7b and 7c figure but placed correctly in the revised manuscript.

Lines 293-294: Comment on "During nighttime and morning, these processes were revered, leading to CO2 storage". During the morning transition the CO2 storage becomes negative (depletion) while during the night storage increases (accumulation). Make this clear instead of just saying "leading to CO2 storage".

✓ Thanks for the comments. It has been clearly explained in the revised manuscript that during nighttime and morning, these processes were reversed, leading to CO2 storage. During the morning transition the CO2 storage becomes negative (depletion) while during the night storage increases (accumulation).

Missed opportunity connecting Figures 7a and 7b: The authors should dig into the inflected results between CO2 storage (7a) and latent energy storage (7b) that occur between 0630 UTC and 0900 UTC during the hours following sunrise. Note the importance of both CO2 and water to photosynthesis which activates at the time where the inflected behavior is observed.

✓ Soil microorganism activities and respiration from soil-pant surface increase over the sunrise as temperature increases, resulting in $CO_2$ release and accumulation near the soil surface and within the canopy which leads to observed $CO_2$ storage (Raza et al., 2022; Davidson and Janssens, 2006; Ryan and Law, 2005). More importantly, mostly at nighttime a stable boundary layer forms due to calm wind conditions which traps the $CO_2$. At sunrise due to temperature rise, the layer begins to weaken. The trapped $CO_2$ mixes with air, disperses, and decreases $CO_2$ storage on sunrise (Stull, 2012).
The higher latent heat storage is due to the latent heat of vaporization, sunlight first hits the dews on the surface and initiates the evaporation which leads to greater latent heat flux, and water vapor stored in the atmosphere leads to higher latent heat storage (Jacobs and Nieveen, 1995).

Raza, T., Qadir, M.F., Khan, K.S., Eash, N.S., Yousuf, M., Chatterjee, S., Manzoor, R., ur Rehman, S. and Oetting, J.N., 2023. Unrevealing the potential of microbes in decomposition of organic matter and release of carbon in the ecosystem. *Journal of Environmental Management*, *344*, p.118529.

Davidson, E. A., & Janssens, I. A. (2006). Temperature sensitivity of soil carbon decomposition and feedbacks to climate change. *Nature*, 440(7081), 165-173. doi:10.1038/nature04514.

Ryan, M. G., & Law, B. E. (2005). Interpreting, measuring, and modeling soil respiration. *Biogeochemistry*, 73(1), 3-27. doi:10.1007/s10533-004-5167-7

Jacobs, A.F. and Nieveen, J.P., 1995. Formation of dew and the drying process within crop canopies. *Meteorological Applications*, *2*(3), pp.249-256.

Stull, R.B., 2012. *An introduction to boundary layer meteorology* (Vol. 13). Springer Science & Business Media.

Line 298: change "After that, this" to "Afterwards the"

✓ "After that, this" has been changed to "Afterwards the" in the revised manuscript.

Lines 299-300: "returning nearly to zero". Its not actually zero

✓ We agree with the reviewer that the values do not actually return to zero, but instead revert to positive (accumulation) values. We have made the necessary clarification and updated the wording accordingly in the revised manuscript.

Line 305: Insert "afternoon into" between "the" and "late nighttime

✓ "afternoon into" has been inserted between "the" and "late nighttime" in the revised manuscript.

Figure 7: There is not a comment about how the data is being processed to form the plots. Is the data vertically integrated? I'm guessing not given the units. If so, what height are the authors choosing or is vertical averaging done? Y-axis label in Figure 7b has a parenthesis that shouldn't be there. Please place the a-c labeling in a corner and not over the plots.

✓ The data processing involved averaging the raw data into 15-minute intervals. The data were analyzed using EddyPro 4.1 software, which processes the raw data by applying block averaging and filtering out outliers based on Reynolds decomposition. Additionally, statistical analysis was performed in R Studio to generate the plots that depict the variation. This approach ensures accurate representation and analysis of the storage terms. Yes, the data are vertically integrated in this instance. A parenthesis in fig 7b has been removed and labeled has been placed in the corner of the graph in the revised manuscript.

Line 313: Change from "the accurate estimation of flux" to "the accurate estimation of fluxes"

✓ "the accurate estimation of flux" has been changed to "the accurate estimation of fluxes".

Figure 7a: How do you explain the large variation between negative and positive CO2 at night and virtually no storage/no variability during the day?

✓ During the morning to onward, transition of the $CO_2$ storage becomes negative (depletion) while during the night storage increases (accumulation). Soil microorganism activities and respiration from the soil-plant interface increase over the

sunrise as temperature increases, resulting in $CO_2$ release and accumulation near the soil surface and within the canopy which leads to observed $CO_2$ storage (Raza et al., 2022; Davidson and Janssens, 2006; Ryan and Law, 2005). More importantly, mostly at nighttime a stable boundary layer forms which traps the $CO_2$. At sunrise due to temperature rise, the layer begins to weaken. The trapped $CO_2$ mixes with air, disperses, and decreases $CO_2$ storage on sunrise (Stull, 2012).This section has been added to the revised manuscript.

Figure quality needs general improvement. Larger fonts, particularly in Figures 5-7 is needed.

✓ The figure quality has been improved in the revised manuscript, and larger fonts have been applied, particularly in Figures 5-7, to enhance readability.

**Reviewer #2**

We have eliminated much of the discussion that would be better presented separately, as a component of a research paper describing the 2023 field experiment, its data analysis and its conclusions. Nevertheless, we offer responses to the Reviewer's comments as follow.

**Major Comments**

the presentation of Fig. 2, especially regarding the transit/sampling time, needs further clarification. The first question is: should we expect the same pattern for intake 2 as shown for intake 1? I do see the first few seconds data of intake 1 should be discarded while the remaining data are stable and good for analysis. But where are the stable, equilibrium regions for intakes #2 and #4? I presume that a similar stable region and the same amount of data points are desired for each height/intake.

✓ Yes, not only 1 and 2 but all intake tubes have the same sampling time. After careful consideration, we decided to remove Table 1 (IRGA output for the multiport air sampling system, for CO2 conc. (430 ppm) fed through the sampling lines at ambient pressure) and Figure 2 (The time-dependent relationship between the infrared gas analyzer (IRGA) in the multiport air sampling apparatus for a gas concentration of 430 ppm $CO_2$ flowing at <1 L/min) from the manuscript, as it raised additional questions and complexities not central to the primary objectives of our study. The initial purpose of this figure was to demonstrate that the instrument was accurately measuring a known $CO_2$ concentration (430 ppm) using different intake tubes, which was a verification step supporting the field trial described in the text that follows. In the revised manuscript, we have clarified that both $N_2$ (0 ppm $CO_2$) and $CO_2$ (430 ppm) were used to test the stability of the system and confirm that the analyzer and switching mechanism were functioning correctly. We believe that the removal of this figure enhances the clarity and focus of the manuscript.

The second question is: at line 181, the authors mentioned that the 4.9 s delay time was improved to 3.2s but did not tell how this was achieved. Did the authors either increase the flow rate or change the tube length? Also, why not show a figure with 3.2 seconds delay since it's the most recent and reasonable result? Using 4.9 seconds delay is confusing here.

✓ We made a mistake. In fact, actual timing for measurement was 3.1 seconds and 4.4 seconds were ignored during the $CO_2$ reading by the analyzer. This has been briefly explained in the revised manuscript.

2) This concern is about section 3.1 Experimental Site. First, the authors started this section with introducing their field instruments and setup, then move to a subsection talking about the basics about the field site. Would it be logically better to introduce the experimental site first, then describe the field measurement setup?

✓ We agree that discussion of the experiment site should be first and the field measurement setup second. In the revised manuscript we have incorporated the reviewer's suggestion

Second, should subsection 3.2 belong to Methodology? It's odd to put it under the section Field Measurement Setup.

✓ Thank you for bringing this to our attention. We have moved subsection 3.2 under the result section.

3) Another concern is about section 4. First, starting at line 247, this paragraph delivers almost the same message as the paragraph above and adds nothing new. Second, starting at line 295, Fig.7b and 7c are mixed up in the figure and in the text, making it hard for readers to follow. The rest of this section lacks further discussion on the results, such as the advantages/limitations of this system compared to other similar studies/systems, a more comprehensive description on what aspects this system can be useful for in future micromet studies.

✓ We have addressed the concerns you raised in the revised manuscript. We have removed the reputation you mentioned in the revised manuscript. We have also fixed the figure 7 a, b & c in the revised manuscript. We have given a complete paragraph describing how this study can be useful for future micromet studies at the end of the result section. Similarly, the limitation of the study has been given in the conclusion and summary. See the revised manuscript.

**Minor comments are also listed below:**

Highlights: the second and third bullet points convey the same meaning: neglecting the storage terms leads to inaccuracies (point 2) while considering it leads to an improved accuracy (point 3). In addition, how this consideration of storage fluxes improved the accuracy of energy closure was not shown in the study. Full budget calculations with/without the storage terms are needed to prove that.

✓  We have changed the text to reflect the purposes of our presentation better.

2) Line 53, the previous two references (Lamas Galdo et al., 2021 and Wang et al., 2023) are very recent, can you find a more recent one to replace this 2004 paper?

✓  Recent references have been added in the revised manuscript (e.g., Hoeltgebaum and Nelson, 2023).

3) Line 56, same as above. Replace this one with a more recent paper if possible.

✓  Recent reference (Varmaghani et al., 2016) has been added in the revised manuscript

4) Line 67, is there a newer paper discussing how many sites are measuring storage? This number must have increased since 2006.

✓  We have updated the number of sites in the revised manuscript and also provided recent references (Fluxnet; Pastorello et al., 2020).

Line 108, a period is missing before 'These'.

✓  A period has been added before "these" in the revised manuscript.

Line 115, the first 'and' changes to 'as'.

✓  Thak you for your attention. "as" has been replaced with "and" in the revised manuscript.

Line 115, 'To minimize consequences of individual sensor offsets". My comment: Actually you avoided the consequence of inconsistency in sensor offsets because you were using a single sensor rather than multiple, so I think you could just say "avoid" instead of "minimize" because it was no longer an issue.

✓  Thank you for your comments. "minimize" has been replaced with "avoid" in the revised manuscript.

Line 162, '3.2 seconds were ignored' changes to 'The first 3.2 seconds of data were discarded, and the remaining ...'

✓  We made a mistake.  In fact, actual timing for measurement was 3.1 seconds and 4.4 seconds were ignored during the $CO_2$ reading by the analyzer. This has been briefly explained in the revised manuscript.

9) Line 162, maybe I missed that but I don't think you have mentioned the frequency of your data so far. Is it 10Hz, 20Hz or else? Since you start to talk about data here, readers may wonder how many data points are there in 4.3 seconds.

✓  The frequency of data was 5Hz which is mentioned in Section 2.1 --  Apparatus design and operation. After careful consideration, we decided to remove Table 1 (IRGA output for the multiport air sampling system, for CO2 conc. (430 ppm) fed through the sampling lines at ambient pressure) and Figure 2 (The time-dependent relationship between the

infrared gas analyzer (IRGA) in the multiport air sampling apparatus for a gas concentration of 430 ppm $CO_2$ flowing at <1 L/min) from the manuscript, as it raised additional questions and complexities not central to the primary objectives of our study. The initial purpose of this figure was to demonstrate that the instrument was accurately measuring a known $CO_2$ concentration (430 ppm) using different intake tubes, which was a verification step supporting the field trial described in the text that follows. In the revised manuscript, we have clarified that both $N_2$ (0 ppm $CO_2$) and $CO_2$ (430 ppm) were used to test the stability of the system and confirm that the analyzer and switching mechanism were functioning correctly. We believe that the removal of this figure enhances the clarity and focus of the manuscript.

10) Line 164, I don't get this 1.8 seconds. You just said 3.2 seconds is the delay time but now you have another delay time. Is the 3.2 seconds delay for the entire switching system and the 1.8 s delay for the IRGA separately? If you did not use this 1.8s delay time for your flux data calculation, I think it would be better to not mention it to avoid confusion.

✓ This part of our text has been thoroughly edited.

11) I don't see any results about the environmental data such as precipitation, temperature, wind, etc. Such data could provide useful information or at least context about this specific location.

✓ A new table 1 is added where precipitation, temperature and storage terms data are provided along the cop height and growth stage. For wind speed see the figure 4. Both table 1 and figure 4 are added into the revised manuscript.

12) Line 211, 'A)' for what?

✓ Thank you for pointing out the mistake. "A)" has been deleted in the revised manuscript.

13) Line 213, font size changed.

✓ The front size has been fixed in the revised manuscript.

14) Line 260, times of sunrise and sunset are NOT shown.

Good catch.

15) Line 262, '15-minute'. My comment: this is the only place you mentioned the average time for flux calculation. You should do this earlier in the method section.

✓ The revision should handle this to your satisfaction.

16) Fig. 5 and Fig. 6: Need to provide a separate table/graph showing how the heights of the intakes were adjusted during the two-month study period (at least in the supplementary materials). Moreover, the sampling heights in these figures are not consistent with what the authors described earlier. For instance, in the figures, the maximum height for the second level is 1.4 m (0.5h), which means max canopy height h is 2.8 m. Then the max

height for the third level (1+h) should be 3.8 m and the max for the top level (2+h) should be 4.8 m. However, in either figure the maximum heights are only 3 m for the third level and 4.4 m for the fourth level. This inconsistency needs explanation.

✓ The instrument evaluation study was conducted for six weeks from May to end of June. The maximum height for H1, H2, H3, and H4 were 0.11m, 1.27m, 3m and 4.36m respectively. In the revised manuscript the measurements along the height and growth stage are provided.

17) Line 303, there were both positive and negative storage before 2000LT.

✓ Latent heat storage (Fig. 7c) fluctuated about zero for most of the daylight hours, after exhibiting a major positive excursion (> 4 W m$^{-2}$) during the few hours after sunrise. After about 2100 LT, similar fluctuations occurred until sunrise, with an average of about –0.5 W m$^{-2}$. We have explained this in the revised manuscript.

18) Fig.7: figure letters a-c need to be placed at the same corner.

✓ We apologize for misplaced letter a-c in fig 7 but placed correctly in the same corner in the revised manuscript

Overall, in my opinion, this paper is not ready yet for publication until major revisions are done.

**Reviewer # 3**

This manuscript (egusphere-2024-2041) deals with the use of a new multiport vertical profile measurement system to examine energy balance closure in an agricultural environment. It is conventionally assumed that storage in the canopy is negligible compared with the other terms, and the procedures for closing the energy balance are not completely standardised.

The methodology used is appropriate for identifying and quantifying this storage term. It aims to respond to the usual technical difficulties (loss of data, multiple devices not operating over common periods, or multiple sensor drifts/offsets, etc.). The system implemented is therefore based on a single device that simultaneously measures temperature, water vapour and CO2. The sampling system therefore enables these variables to be analysed at several heights over short time steps (a few seconds).

1. On the pure technical elements, it is clear that the analysis for each of the tubes merits the exclusion of residual air, estimated at 3.2 seconds, but it is not clear how this calculation of the purge delay is made. You could better explain this section.

   ✓ Air sampling through the system is controlled by the four, 3-way solenoid valves. Data logger controls the switching of the valves that switches the air between the sampling pump and purging pump sequentially. The sampling pump draws air from the height of interest and passes it to the analyzer (IRGA) for 7.5

seconds. The first 4.4 seconds of data allows the logger to equilibrate and the next 3.1 seconds with the reading of the sample. The delay of 4.4 seconds before the reading of the data logger is based on the equilibration of the system and purging of air from a separate height. This has been briefly explained in the revised manuscript.

2. Regarding figure 2, I didn't understand the change from 4.9 seconds to 3.2 seconds. You could better explain this section.
   ✓ We made a mistake. In fact, actual timing for measurement was 3.1 seconds and 4.4 seconds were ignored during the $CO_2$ reading by the analyzer. This has been briefly explained in the revised manuscript.

3. Lines 205 and 206 give an average temperature and precipitation, but two values (a range of values) are given. I didn't understand the reason for this two averages. In my opinion, only one average is possible.
   ✓ We have changed into average temperature and precipitation 13.5 °C and 54 in respectively.

4. Figure 4 doesn't provide much information. I suggest deleting it.
   ✓ We have removed figure 4 in the revised manuscript.

5. In my opinion, there is a lack of information on the maize crop. At the very least, we need information on the sowing and harvest dates, if the ris no infiorration about the average LAI over the period, and ideally the maize variety. It would be important to provide the soil depth that can be explored by the root system to get an idea of the water reserve (large or small) available to the plant (and therefore potential water stress).
   ✓ The current study was for 6 weeks from growth stage V2 to growth stage VT (last week of April 2023 to last week of June 2023). We have not recorded the LAI till July 2023. The maize variety was "Dekalb 66-06". We have not measured the root systems but in literature it is reported that maize root length varies between 1 and 1.5 meters.

6. Figures 5 and 6 could be improved (font size, box, grid, etc.).
   ✓ Figures 5 and 6 have been improved according to reviewer suggestion.

7. I do not recommend using "approximately" or "around" (line 298), which are not « countable ». I suggest giving a range or an average value or any other indication that is not open to interpretation by the reader.
   ✓ We have replaced it with an average value of 2 W m$^{-2}$

8. Figures 5, 6 and 7 show envelope curves. What does the envelope correspond to? Unless I'm mistaken, I don't have the information. I think it's important to mention this in the legends so that each of these figures can stand on its own.
   ✓ We have changed the figure 5, 6, and 7 in the revised manuscript. In the revised figures, symbols correspond to different heights of measurements with bands corresponding to +/- one standard error.

Despite these comments, many of which are formal, I find this new system interesting and relevant for such energy storage estimates.

Thank you for your thoughtful feedback and for finding our system interesting and relevant for energy storage estimates. We greatly appreciate your recognition of the potential of our work.

We have carefully considered all the comments you provided and have addressed them in the revised manuscript.

---

## Author Response (AR2)

**Date:** Tuesday, July 29, 2025

To,

Dr. Russell Dickerson
Handling Editor
**Atmospheric Measurement Techniques**

Subject   Submission minor revision of research paper **egusphere-2024-2041** to Atmospheric Measurement Techniques

Dear Sir,

We are resubmitting herewith minor version of our research paper on **"A simplified system to quantify storage of carbon dioxide, water vapor and heat within a maize canopy (egusphere-2024-2041)".**  We are grateful to the reviewers for their careful and helpful analysis of our manuscript. Based on reviewers' comments, the manuscript has been substantially improved.

Thank you for receiving our manuscript and considering it for publication. The authors appreciate your valuable time and look forward to your response.

Sincerely yours,
Corresponding author
Taqi Raza
taqiraza85@gmail.com, traza@vols.utk.edu
Department of Biosystems Engineering & Soil Science
University of Tennessee, Knoxville USA

Minor comments

"First, we must emphasize our regret that we failed to explain the reason for devising the
sampling system now described. This has now been rectified, and we hope that some of the Reviewer's misgivings will be assuaged."

Great – but the Abstract needs to make this point and stronger in the Introduction. "The focus is on small farms in complex terrain, such as dominating most of Africa and much of eastern North America. "

✓  The point has been addressed in the abstract in the revised manuscript.

Put all figures immediately after first call out.

✓  All the figures have been put after first call out.

In Conclusions there is a big difference between "few disruptions" and "A few disruptions." Which do you mean?

✓  A few disruption

And again, make your point about how this simplified equipment is suitable for remote locations!

✓  We have explained it in conclusion.

Your Table 1 cannot be split into two "panels". Please use a single table and rotate it by 90 degrees if it does not fit on a portrait format page.

✓  Table 1 has been formatted according to this suggestion.